



# High-resolution land-use land-cover change data for regional climate simulations over Europe - Part I: The plant functional type basemap for 2015

Vanessa Reinhart[1,2], Peter Hoffmann[1,2], Diana Rechid[1], Jürgen Böhner[3], and Benjamin Bechtel[4]

[1]Helmholtz-Zentrum Hereon, Climate Service Center Germany (GERICS), Fischertwiete 1, 20095 Hamburg, Germany
[2]Universität Hamburg, Institut of Geography, Section Physical Geography, Bundesstraße 55, 20146 Hamburg, Germany
[3]Universität Hamburg, Institute of Geography, Cluster of Excellence "Climate, Climatic Change, and Society" (CLICCS), 805A Geomatikum, Bundesstraße 55, D-20146, Hamburg, Germany
[4]Ruhr-Universität Bochum, Department of Geography, Universitätsstraße 150/ Gebäude IA, 44801 Bochum, Germany

**Correspondence:** Vanessa Reinhart (vanessa.reinhart@hereon.de)

**Abstract.** The concept of plant functional types (PFTs) is shown to be beneficial in representing the complexity of plant characteristics in land use and climate change studies using regional climate models (RCMs). By representing land use and land cover (LULC) as functional traits, responses and effects of specific plant communities can be directly coupled to the lowest atmospheric layers. To meet the requirements of RCMs for realistic LULC distribution, we developed a PFT dataset 5 for Europe (LANDMATE PFT Version 1.0; Reinhart et al., 2021b). The dataset is based on the high-resolution ESA-CCI land cover dataset and is further improved through the the additional use of climate information. Within the LANDMATE PFT dataset, satellite-based LULC information and climate data are combined to achieve the best possible representation of the diverse plant communities and their functions in the respective regional ecosystems while keeping the dataset most flexible for application in RCMs. Each LULC class of ESA-CCI is translated into PFT or PFT fractions including climate information by 10 using the Holdridge Life Zone concept. Through the consideration of regional climate data, the resulting PFT map for Europe is regionally customized. A thorough evaluation of the LANDMATE PFT dataset is done using a comprehensive ground truth database over the European Continent. A suitable evaluation method has been developed and applied to assess the quality of the new PFT dataset. The assessment shows that the dominant LULC groups, cropland and woodland, are well represented within the dataset while uncertainties are found for some less represented LULC groups. The LANDMATE PFT dataset provides a 15 realistic, high-resolution LULC distribution for implementation in RCMs and is used as basis for the LUCAS LUC dataset introduced in the companion paper by Hoffmann et al. (submitted) which is available for use as LULC change input for RCM experiment setups focused on investigating LULC change impact.





## 1 Introduction

Land use and land cover (LULC), including the vegetation type and function, was declared an Essential Climate Variables (ECVs) by the Global Climate Observing System (GCOS) (Bojinski et al., 2014). Changes in ECVs are crucial factors of climate change and therefore need to be monitored and further represented in climate models to be able to assimilate and understand atmospheric processes and feedback effects on different scales. For LULC, anthropogenic modifications are the most important drivers of change. De- and reforestaion and expansion of urban and cropland areas affect biogeophysical (e.g.,

albedo, roughness, evapotranspiration, runoff) and biogeochemical (e.g., carbon emissions and sinks) surface properties and processes (Mahmood et al., 2014; Lawrence and Vandecar, 2015; Alkama and Cescatti, 2016; Perugini et al., 2017; Davin et al., 2020). Besides LULC changes, land management practices are being assessed regarding influence of related land surface modifications on regional climate, and also the potential of land management practices regarding climate change adaptation and mitigation efforts (Lobell et al., 2006; Kueppers et al., 2007; Burke and Emerick, 2016).

In order to represent impacts and feedbacks of LULC modifications as realistic as possible, regional climate models (RCMs) require an accurate representation of LULC and its changes. In this context, the concept of plant functional types (PFTs) is increasingly used for the representation of LULC in RCMs. A comprehensive review of the subsequent development of PFTs representing vegetation dynamics in climate models was done by Wullschleger et al. (2014). The need for applicable global PFT maps for vegetation models that are used with atmospheric models was already well emphasized by Box (1996).

Moreover, the requirement that a climate model should include a vegetation model representing the biosphere was discussed by Lavorel et al. (2007). One criterion that is highly emphasized is the inter-regional applicability of a preferably simple PFT classification, which has the ability to capture key characteristics of the biosphere from biome to continental scale, regardless of climate zone and individual vegetation composition. A variety of PFT definitions and cross-walking procedures (CWPs), used for translating LULC products into global or regional PFT maps, are currently available. The European Space Agency

Climate Change Initiative (ESA-CCI) and the United States Geological Service (USGS) provide the only two ready to use continuous global products to the community (Poulter et al., 2015; Sulla-Menashe and Friedl, 2018). However, the individual PFT definitions and CWPs as well as the mostly satellite based input data differ greatly in complexity and temporal and horizontal resolution (Bonan et al., 2002; Winter et al., 2009; Lu and Kueppers, 2012). Moreover, inter-regional consistency cannot be achieved by products that origin from regionally constrained input data or regionally adapted CWPs. Therefore,

the additional use of climate information in the CWP from LULC to PFT is a highly useful step, to create a dynamically customizable product, that can be adapted to various climate and vegetation characteristics (Poulter et al., 2011).

With the present work, we introduce a PFT map for the European Continent that specifically addresses the requirements of the RCM community (Bontemps et al., 2013). The land cover maps of the ESA-CCI are translated into 16 PFTs creating an updated version of the interactive MOsaic-based Vegetation (iMOVE) PFTs that were originally developed for the RCM REMO

(Wilhelm et al., 2014). Climate information is implemented into the CWP employing the Holdridge ecosystem classification concept based on the Holdridge Life Zones (HLZs; Holdridge et al., 1967), which provide a global classification of climatic zones in relation to potential vegetation cover. The HLZ concept is commonly used as a tool for ecosystem mapping from



various overlapping research communities (Lugo et al., 1999; Yue et al., 2001; Khatun et al., 2013; Szelepcsényi et al., 2014; Tatli and Dalfes, 2021). This paper gives a detailed documentation on the preparation of the PFT map - hereinafter referred

to as "LANDMATE PFT" - within the Helmholtz Institute for Climate Service Science (HICSS) project "Modelling human LAND surface Modifications and its feedbacks on local and regional cliMATE" (LANDMATE). The LANDMATE PFT map is prepared in close collaboration with the EURO-CORDEX Flagship Pilot Study Land Use and Climate Across Scales (FPS LUCAS; Rechid et al., 2017). Within the FPS LUCAS, RCM experiments are coordinated among an RCM ensemble to investigate the impact of LULC change for past climate and future climate scenarios. Through creation of LANDMATE PFT

and the time series LUCAS LUC (Hoffmann et al., submitted), the need for improved LULC and LULC change representation among the FPS LUCAS RCM ensemble is met. For the preparation of LANDMATE PFT, we developed a CWP for the translation of LULC classes of ESA-CCI into 16 PFTs according to the needs of regional climate modellers from all over Europe (Bontemps et al., 2013). A key issue to address in the map development process is the accuracy of LULC representation in the final product (Hartley et al., 2017). In order to assess the quality of the product, we compared the LANDMATE PFT

map to a comprehensive ground truth database for large parts of the European Continent. The quality information derived from the assessment supports the RCM community in addressing and interpreting uncertainties caused by LULC representation in RCMs. The general workflow and subsequently all utilized datasets are summarized in section 2 while the major steps of the CWP are listed in section 3. Section 4 introduces in detail the accuracy assessment procedure followed directly by the results in section 4.3. All CWTs and figures corresponding to the CWP and the accuracy assessment can be found in Appendix A and

B.

## 2 Methods and data

The LANDMATE PFT map (Reinhart et al., 2021b) is a combination of multiple datasets and concepts created using well-established methods and in addition, by considering the expertise of regional climate modellers from all over Europe within the FPS LUCAS.

### 2.1 General workflow

The workflow to generate the LANDMATE PFT map is summarized in fig. 1, which also includes the steps to generated the LUCAS LUC dataset further described in the companion paper by Hoffmann et al. (submitted). First, the ESA-CCI land cover map (Sect. 2.2.1), which has a native resolution of ~300 m, is aggregated to the 0.1° target resolution using SAGA GIS (Conrad et al., 2015). The target resolution results from the FPS LUCAS ensemble resolution (i.e., EURO-CORDEX domain EUR-11)

that is used for LULC change impact studies in FPS LUCAS Phase II. The LULC type information from the original product is preserved in fractions per 0.1° grid cell which is advantageous to common majority resampling methods. The sum of PFT fractions in the whole dataset remains the same in all target resolutions, only the distribution of fractions per grid cell changes depending on the target resolution.

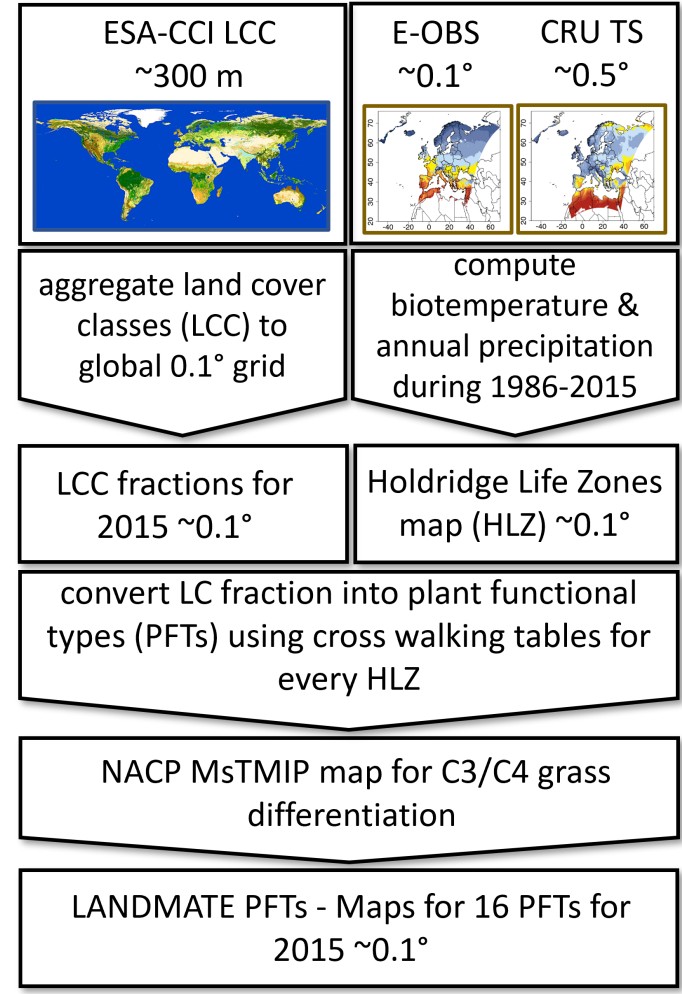

**Figure 1.** The general workflow to generate LANDMATE PFT 2015 Version 1.0. This workflow is part of the workflow to generate the LUCAS LUC time series as introduced in the companion paper by Hoffmann et al. (submitted)

The E-OBS gridded climate data (Sect. 2.2.2) is utilized for the preparation of the HLZ map over Europe (Sect. 2.2.4). From
E-OBS, the ensemble mean 2-meter-temperature and annual precipitation from 1950-2020 are used to create the HLZ map of
0.1° horizontal resolution which is further implemented in the CWTs to prepare the final LANDMATE PFT maps. For regions
that are not covered by E-OBS, the respective data of the CRU dataset (Sect. 2.2.3) is used.

For each of the 37 ESA-CCI land cover classes, an individual CWT is created (Sect. 3) that includes a unique translation
for each used HLZ. The translation process is based on Wilhelm et al. (2014) where the translation of the Global Land Cover
(GLC) 2006 to the 16 REMO-iMOVE PFTs is described. Since the nomenclature of GLC 2006 and ESA-CCI LC are similar
and based on the same classification system some of the CWTs were initially adopted from (Wilhelm et al., 2014). For the
more diverse ESA-CCI LC classes new CWTs need to be created. The new CWTs follow the translation of Poulter et al.



(2015) (ESA-CCI PFTs) but were carefully revised and modified during the process. This revision of the CWTs is supported
by reference data and visual satellite image interpretation. The quality of the LANDMATE PFT dataset is finally assessed by
comparison to a comprehensive ground truth database (Sect. 4).

## 2.2 Datasets & concepts

### 2.2.1 ESA-CCI LC

The European Space Agency Climate Change Initiative (ESA-CCI) provides continuous global land cover maps (ESA-CCI
LC) on ~300 m horizontal grid resolution. The ESA-CCI LC maps are available for download in annual time steps for the
100   years 1992-2018 (ESA, 2017). The classification of the LC maps follows the United Nations Land Cover Classification System
(UN-LCCS) protocol (Di Gregorio, 2005) and consists of 22 level 1 classes and 14 additional level 2 classes, which include
regional specifications. More information on ESA-CCI LC data processing can be found at maps.elie.ucl.ac.be/CCI/viewer/
download/ESACCI-LC-Ph2-PUGv2_2.0.pdf. An overview of the satellite missions involved in the production of ESA-CCI
LC is given in table 1. Besides systematic global validation efforts (ESA, 2017; Hua et al., 2018), a few regional approaches
investigated the quality of ESA-CCI LC over Europe (Vilar et al., 2019; Reinhart et al., 2021a).

**Table 1.** Satellite missions involved in the production of ESA-CCI LC according to ESA (2017)

| Time period | Satellite product |
|---|---|
| Baseline Production 2003-2012 | MERIS FR/RR [1] global SR [2] composites |
| 1992-1999 | Baseline 10-year global map; AVHRR [3] global SR composites for back-dating baseline |
| 1999-2013 | Baseline 10-year global map; SPOT-VGT [4] global SR composites for up and back-dating the baseline; PROBA-V [5] global SR composites at 300 m |
| 2013-2015 | Baseline 10-year global map; PROBA-V global SR composites at 1 km for years 2014 and 2015 for up-dating the baseline; PROBA-V time series at 300 m |
| Since 2016 | Sentinel-3 OLCI and SLSTR [6] 7-day composites |

[1]MEdium Resolution Imaging Spectrometer Full Resolution/Reduced Resolution (ESA, 2002)
[2]Surface Reflectance
[3]Advanced Very-High-Resolution Radiometer (Hastings and Emery, 1992)
[4]SPOT Vegetation satellite program (Maisongrande et al., 2004)
[5]Project for On-Board Autonomy - Vegetation (Dierckx et al., 2014)
[6]Ocean and Land Colour Instrument (OLCI) and Sea and Land Surface Temperature Radiometer (SLSTR) (Donlon et al., 2012)



### 2.2.2 E-OBS Climate data

The E-OBS dataset (Cornes et al., 2018) is a daily gridded observational dataset, derived from station observations from European countries covering the period from 1950 to 2020. The point observations are interpolated using a spline method with random perturbations in order to produce an ensemble of realizations. For the creation of the HLZs that are used for the conversion of ESA-CCI LC classes to PFTs (Section 2.2.5), the ensemble mean of the 2-meter-temperature (TG) and precipitation (RR) on a regular 0.1° grid from E-OBS version 19.0e is used. It covers most of Europe, some parts of the Middle East and a narrow strip of Northern Africa.

### 2.2.3 CRU

The Climate Research Unit (CRU) TS 4.03 dataset is a global gridded high-resolution climate dataset based on station observations produced and maintained by the CRU of the University of East Anglia (Harris et al., 2014). The dataset provides global monthly means of climate parameters at 0.5° resolution from 1901 to 2019. In order to achieve the target resolution of 0.1° for the global LANDMATE PFT maps, the CRU climate data is downscaled using bilinear interpolation. Following Hoffmann et al. (2016), distance-weighted interpolation was applied to the atmospheric observation dataset CRU to extrapolate the climate data to the coastlines of the ESA-CCI LC maps in order to compensate for the different land-sea-masks of the products. The CRU climate dataset was used within this application for regions where E-OBS is not available.

### 2.2.4 Holdridge Life Zones

The Holdridge Life Zone (HLZ) concept was initially developed in 1967 (Holdridge et al., 1967) to define all divisions of the global biosphere, depending on the relation of biotemperature (average of monthly temperature above 0°C; since plant activities are idle below freezing, all values below 0°C are adjusted to 0°C), mean annual precipitation and ratio of potential evapotranspiration to mean annual precipitation. By combining threshold values of biotemperature and annual rainfall, the 38 HLZs are created (Table 2). In the present analysis, the tropical and subtropical as well as the polar and subpolar HLZs are mereged. Through the merging of the aforementioned HLZs, 30 individual HLZs in total are available for the creation of the European HLZ map (Fig. 2). The dynamic character of the specific quantitative ranges of the long-term means of the

**Table 2.** The Holdridge Life Zones following (Holdridge et al., 1967).

| Bio-temperature [°C] | Precipitation [mm] | | | | | |
|---|---|---|---|---|---|---|
| | <125 | 125 to <250 | 250 to <500 | 500 to <1000 | 1000 to <2000 | >2000 |
| <3 | Subpolar dry tundra | Subpolar moist tundra | Subpolar wet tundra | Subpolar rain tundra | - | - |
| 3 to <6 | Boreal desert | Boreal dry shrub | Boreal moist forest | Boreal wet forest | Boreal rain forest | - |
| 6 to <12 | Cool temperate desert | Cool temperate desert shrub | Cool temperate steppe | Cool temperate moist forest | Cool temperate wet forest | Cool temperate rain forest |
| 12 to <18 | Warm temperate desert | Warm temperate desert scrub | Warm temperate thorn steppe/woodland | Warm temperate dry forest | Warm temperate moist forest | Warm temperate wet/rain forest |
| 18 to <24 | Subtropical desert | Subtropical desert shrub | Subtropical thorny steppe/woodland | Subtropical dry forest | Subtropical moist forest | Subtropical wet/rain forest |
| >24 | Tropical desert | Tropical desert shrub | Tropical thorny woodland | Tropical very dry forest | Tropical dry forest | Tropical moist/wet/rain forest |

utilized climate parameters make the HLZ classification more flexible than other available global ecosystem classifications and



therefore makes the HLZs most suitable for the application presented in this article. In addition the requirement for input data is relatively low.

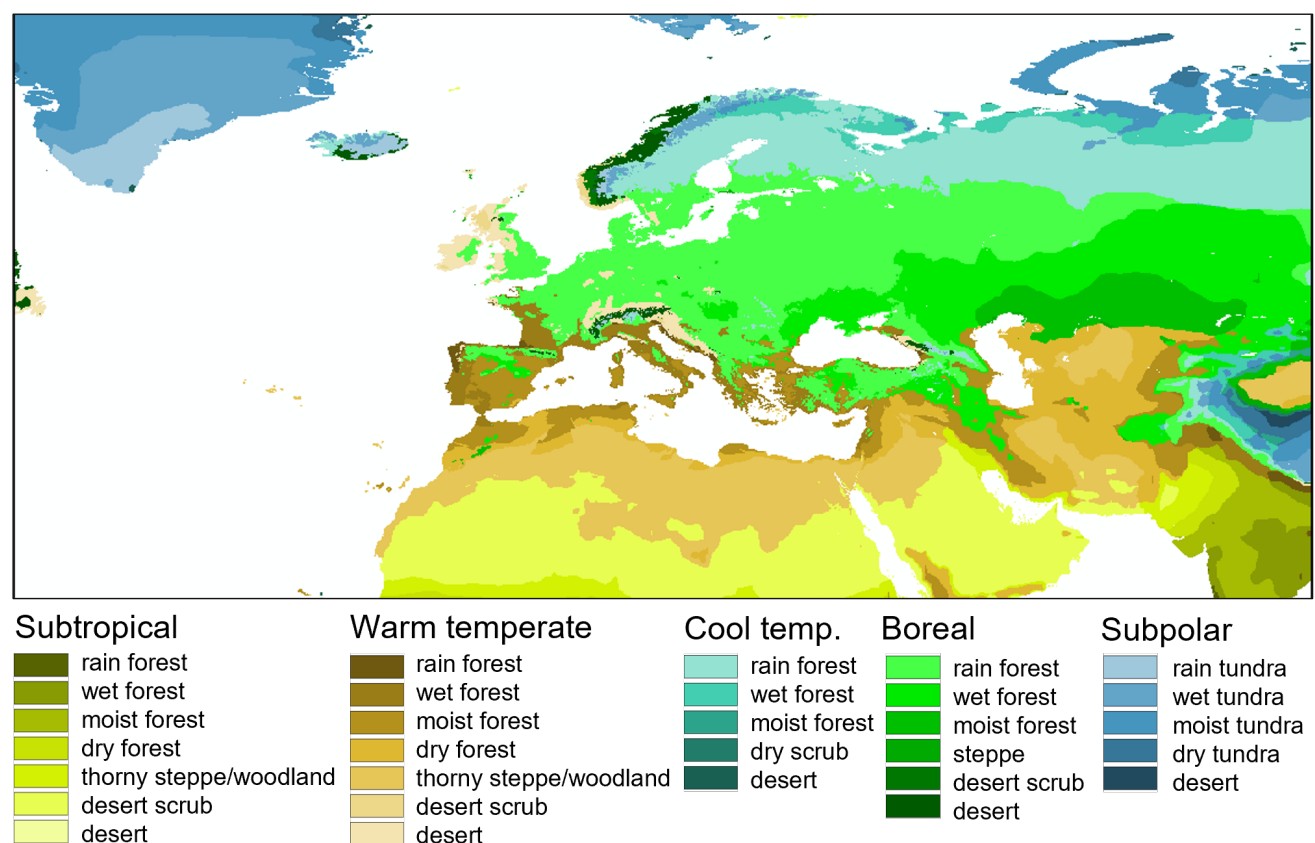

**Figure 2.** Holdridge Life Zones map for the extent of LANDMATE PFT

In the past, the HLZ concept was not only found useful for global applications but successfully implemented especially for regional mapping approaches due to its ability to capture regional climate features with the support of bioclimatic variables (Daly et al., 2003; Tatli and Dalfes, 2016). Further, the HLZ concept was used for LULC change predictions, such as land use

impact assessments, related to current and future climate change scenarios (Chen et al., 2003; Skov and Svenning, 2004; Yue et al., 2006; Saad et al., 2013; Szelepcsényi et al., 2018). With the implementation of climate data through the HLZ concept, the resulting PFT maps become more detailed and can be customized to individual regions without losing global consistency.

### 2.2.5 Plant Functional Types

Table 3 shows the LANDMATE PFTs that are based on the PFTs introduced by Wilhelm et al. (2014). The implementation of an

irrigated cropland PFT (PFT 14) that is currently being developed within the HICSS project LANDMATE will be implemented





in a later version of the dataset. In the initial version that is presented in this article, all cropland proportions are assigned to the cropland PFT (PFT 13).

**Table 3.** LUCAS plant functional types based on Wilhelm et al. (2014) with modified crop types.

| PFTs | Names |
| --- | --- |
| 1 | Tropical broadleaf evergreen trees |
| 2 | Tropical deciduous trees |
| 3 | Temperate broadleaf evergreen trees |
| 4 | Temperate deciduous trees |
| 5 | Evergreen coniferous trees |
| 6 | Deciduous coniferous trees |
| 7 | Coniferous shrubs |
| 8 | Deciduous shrubs |
| 9 | C3 grass |
| 10 | C4 grass |
| 11 | Tundra |
| 12 | Swamp |
| 13 | Non-irrigated crops |
| 14 | *Irrigated crops* [7] |
| 15 | Urban |
| 16 | Bare |

### 2.2.6 Potential C4 grass fraction NACP MsTMIP

The initial land cover map from the ESA-CCI LC does not provide a distinction between C3 and C4 grassland. Therefore, an
145 additional product is used after applying the CWP. The map from the North American Carbon Program Multi-scale Synthesis and Terrestrial Model Intercomparison Project (NACP MsTMIP; Wei et al., 2014) is constructed based on the synergetic land cover product (SYNMAP) by Jung et al. (2006). SYNMAP is a combination of multiple high-resolution LULC products using a fuzzy agreement approach. The NACP MsTMIP map uses the grassland fractions from the SYNMAP product and the C4 grass distribution is estimated supported by growing season temperature based on present climate conditions (Wei et al., 2014).
The map is provided in 0.5° horizontal grid resolution for the period from 1801 to 2010. For the preparation of LANDMATE PFT the NACP MsTMIP map of 2010 is used.

---

[7]the irrigated crop PFT is currently empty (see section 3.4)





## 2.3 LUCAS - land use and land cover survey

The harmonized LUCAS *in situ* land cover and use database for field surveys from 2006 to 2018 (d'Andrimont et al., 2020) is the most consistent ground truth database for the European Continent. The survey was carried out at three-yearly intervals between 2006 and 2018. The systematic sampling design of the survey consists of a theoretical, regular grid over the European Continent with ~2 km grid size. The reference point locations are the corner points of the theoretical grid. Not all locations within the survey were easily accessible. Therefore, the survey is supported by in situ photo interpretation, in-office photo interpretation and satellite data in the latest time steps 2015 and 2018 (table 4). However, the main proportion of the reference points was recorded through location visits at all time steps, which makes this land survey the most reliable and consistent ground truth database for Europe.

**Table 4.** Number and recording method of reference points in the LUCAS land cover and use database per timestep.

| Year | Reference points | in situ | in situ PI[8] | in-office PI[9] | GT[10] [%] |
|------|------------------|---------|---------------|-----------------|------------|
| 2006 | 168401 | 155238 | 13163 | | 92.18 |
| 2009 | 234623 | 175029 | 59594 | | 74.6 |
| 2012 | 270272 | 243603 | 26669 | | 90.13 |
| 2015 | 340143 | 242823 | 25254 | 71970 | 71.39 |
| 2018 | 337854 | 215120 | 22894 | 99803 | 63.67 |

The extent of the LUCAS survey was increased over time. The 2006 survey covered 11 countries while the 2018 map covers large parts of the European Continent with 28 countries. Throughout the survey, the ground truth data has been continuously checked for quality and plausibility. For the accuracy assessment of the LANDMATE PFT map the ground truth points of the year 2015 are employed (Sect. 4). In order to avoid confusion between the FPS LUCAS and the LUCAS ground truth dataset, the latter will be further referred to as **Ground Truth Survey** or **GT-SUR**.

## 3 Cross-walking procedure - ESA-CCI LC classes to PFTs

The CWP from ESA-CCI LC classes to PFTs presented in this article is based generally on (1) the translation introduced by Poulter et al. (2015) and (2) the translation by Wilhelm et al. (2014). Both translations are not just combined with each other but modified using additional data. The following sections introduce the PFTs of LANDMATE PFT aggregated into groups and give an overview of the decisions on modifications that are made during the production process based on literature and additional data. The final LANDMATE PFT map is shown in fig. 3.

---

[8]Photo interpretation close to the reference location

[9]Photo interpretation with supporting data, such as satellite images

[10]Ground truth





## 3.1 Trees and shrubs, tropical and temperate | PFT 1-8

The LANDMATE PFTs are more diversified regarding tree-PFTs than the generic ESA-CCI PFTs. The expansion of tree-PFTs to six in total was done at the expense of two shrub-PFTs. The increase of tree-PFT diversity is done in order to address the strong biogeophysical impacts of forested areas on regional and local climate, such as decreased albedo and increased roughness length (Bright et al., 2015). The effects of forested areas on near-surface climate are distinctively different to the effects of shrub or grass covered areas, and are also highly depending on tree species composition and latitudinal range (Bonan, 2008; Richardson et al., 2013). Another reason for the six tree-PFTs is the intended use of the PFT maps in RCMs. In the Land Surface Models (LSMs) of current generation RCMs, where a distinction is rather made between different tree or tree community types than between different shrub types. Therefore and with regard to the implementation process that needs to be done for each RCM individually, an increase in the number of tree-PFTs and a decrease in the number of shrub-PFTs is considered to be convenient. Accordingly, the tree and shrub proportions were distributed following both, the needleleaf and broadleaf definitions of the ESA-CCI LC classes as well as the HLZ map, where the HLZ map was decisive for an assignment of forest proportions to the temperate or tropical tree-PFT, respectively. Following a comparison with different forest datasets over Europe (not shown), the tree proportions in the translation of the mixed land cover classes (e.g. lass 61 - Tree cover, broadleaved, deciduous, closed (>40%)) are increased to be in line with the indicated overall forest amount over Europe.

## 3.2 Grassland | PFT 9 & 10

The generic ESA-CCI PFTs include a natural grassland- and a managed grassland-PFT to include grassland and cropland respectively. The LANDMATE PFTs include two grassland-PFTs, distinguishing between C3 and C4 grass. The contrasting photosynthetic pathways and therefore contrasting synthetic response to $CO_2$ and temperature determine specific ecosystem functions for both PFTs respectively. The main differences are found in global terrestrial productivity and water cycling (Lattanzi, 2010; Pau et al., 2013). The translation from the LULC classes that contain grassland proportions into C3 or C4 grass-PFTs respectively is supported by a map of potential C4 vegetation by Wei et al. (2014) where the potential global distribution of C4 is estimated using bioclimatic parameters (Sect. 2.2.6).

## 3.3 Tundra and swamps | PFT 11 & 12

The specific vegetation PFTs tundra and swamps are treated individually in LANDMATE PFT. Tundra is mostly used for the polar and subpolar HLZs, where the climatic conditions require a clear distinction of the land surface properties to the boreal and temperate regions regarding exchange and feedback processes with the atmosphere (Thompson et al., 2004). Chapin Iii et al. (2000) further suggest a differentiation of vegetation composition within these northern vegetation communities, which can also be realized using the introduced translation. The swamp-PFT is mostly used for translating the ESA-CCI LC mosaic tree/shrub/herbaceous classes and also partly for the flooded tree cover classes in most of the HLZs. Swamps occur mainly in the boreal and polar regions.



### 3.4   Cropland | PFT 13 & 14

Currently, two cropland-PFTs are defined in the LANDMATE PFT map. The cropland-PFT (PFT 13, see table 3) includes

all managed, agricultural land surface proportions. The uncertainties of the translation of the ESA-CCI cropland classes and mixed cropland classes into the cropland-PFTs was investigated by Li et al. (2018) where the comparison of LULC change in the ESA-CCI PFT maps against other LULC products showed inconsistencies between global trends and geographical patterns between the products. However, Li et al. (2018) provide a modified CWT that was adjusted in regard to an improved knowledge base on how to translate LULC classes into PFTs for climate models. Particular focus is laid on mosaic classes and the sparsely

vegetated classes of which appear numerous in ESA-CCI LC. Therefore, the translation from Li et al. (2018) for cropland is adopted into the present CWP.

The irrigated cropland-PFT (PFT 14, see table 3) is currently empty in the LANDMATE PFT map Version 1.0. This decision is made following intense research on available irrigation information. The ESA-CCI LC map that is used as initial input contains an "irrigated cropland" class but this information was not used in the process. The investigation on irrigated areas included

the comparison of ESA-CCI LC to other products that are available, such as the irrigation map from the FAO (Siebert et al., 2005).Although the ESA-CCI LC quality assessment shows a very good agreement of the ESA-CCI LC irrigated cropland with the validation database (ESA, 2017), the comparison showed considerable differences between the products. The success of detection of irrigated areas is highly dependent on the correct detection of the crop types to infer the water needs of the respective crops, on atmospheric and environmental conditions and on the availability of multi-temporal, high resolution im-

agery (Bégué et al., 2018; Karthikeyan et al., 2020). Further, most remote sensing applications depend highly on ground truth data and local knowledge. Applications using different satellite imagery to detect agricultural management practices, such as irrigation, are only successfully tested and applied in local spatial units (Rufin et al., 2019; Ottosen et al., 2019). Therefore, the irrigated cropland PFT remains unoccupied for now. Nevertheless, PFT 14 is defined within LANDMATE PFT Version 1.0 for the purpose of adding irrigated LULC fractions in the future. For the long term LUCAS LUC dataset (Hoffmann et al.,

submitted) which is extended backward and forward based on the LANDMATE PFT map for Europe 2015, irrigated cropland areas are already implemented following the irrigated area definition of the Land Use Harmonization (LUH2) dataset (Hurtt et al., 2011).

### 3.5   Non-vegetated | PFT 15 & 16

The non vegetated-PFTs in the LANDMATE PFT dataset are urban and bare. The urban grid cells from ESA-CCI LC are

directly translated into urban fractions for all HLZs in the CWP. The same applies for all bare ground proportions that are translated fully into the bare-PFT. In addition, the ESA-CCI LC mixed classes are split up and the bare ground proportions within the mixed classes are added to the bare-PFT. The explicit treatment of urban areas and especially differentiation from bare ground provides the possibility to resolve urban surface characteristics in RCMs. The treatment of urban areas as a slab surface or as an equal to rock surface as done in several RCM approaches cannot account for the complex geobiophysical





processes associated with an urban agglomeration (Daniel et al., 2019; Belda et al., 2018). Due to the distinction of the two surface types, the LANDMATE PFT map can be used for impact studies with an urban focus.

### 3.6 Water, permanent snow & ice

The LANDMATE PFTs do not include individual PFT definitions for water and snow/ice respectively. Regarding the water representation, most currently used RCMs are utilizing a land-sea-mask to account for oceans and inland water areas. Therefore,

an explicit definition of water as individual PFT has not been implemented. Consequently, water grid cells are set to no data. In the present translation, the snow/ice grid cells from ESA-CCI land cover are translated into bare-PFT following Wilhelm et al. (2014).

**Figure 3.** LANDMATE PFT map for Europe for 2015 (a). Below a map section of the Alpine region shows an example of the resolution difference between LANDMATE PFT 0.1 (b) and LANDMATE PFT 0.018 (c). LANDMATE PFT 0.018 is used in the present accuracy assessment. For improved visualization all maps show the majority PFT per grid cell.





## 4 Quality assessment of the LANDMATE PFT map

The LANDMATE PFT map is based on the ESA-CCI LC map which was quality checked and compared to similar LULC
products on a global (ESA, 2017; Yang et al., 2017; Hua et al., 2018; Li et al., 2018) and regional level (Reinhart et al., 2021a;
Vilar et al., 2019). However, the translation from LULC classes to PFTs necessarily results in change of the map. The final
product, the LANDMATE PFT map, is intended to be used in RCMs, which means the quality of the final product must be
assessed in addition to the available quality assessments of the initial ESA-CCI LC map. In order to overcome the resolution
difference, which is non negligible between LANDMATE PFT and the reference data GT-SUR, the LANDMATE PFT map is
prepared on 0.018° horizontal resolution, which corresponds closely to the 2 km theoretical grid of GT-SUR.

The design of such a quality assessment of a large scale map product is not trivial, especially since the map product itself
and the reference data are often different in structure and nomenclature, given that ground truth reference data is mostly
collected as point data and independently from the assessed map product Foody (2002); Wulder et al. (2006); Olofsson et al.
(2014). In order to produce reliable quality information for LANDMATE PFT, the present assessment follows closely the
well established good-practice recommendations. Nevertheless, adjustments are done to account for the fractional structure of
LANDMATE PFT. Section 4.2 provides additional information on the requirements of a "good practice" accuracy assessment,
the key components and the selected sampling design and metrics.

### 4.1 Research area

The coverage of GT-SUR in the year 2015 includes 28 countries which are highlighted in dark grey in fig. 4.



**Figure 4.** Coverage of the Land Use and Coverage Area frame Survey (LUCAS) for reference year 2015 (top). The lower figure shows the points and LULC group representation within the grid cell highlighted in black color in the top map as an example for the whole research area.





The total number of GT-SUR points for 2015 is 340,143. Out of these points, 338,619 points (~99.55%) are covered with valid LANDMATE PFT grid cells of the assessed LULC groups and can be used in the analysis. Countries located within the contiguous area but missing in the assessment are Switzerland, Norway, the Russian Oblast Kaliningrad, Bosnia and Herzegovina, Montenegro, Albania, Serbia, Kosovo, North Macedonia, and Belarus. Figure 4 also shows the 2.5° grid that was used for the analysis of the accuracy assessment results (Sect. 4.3). Due to the fine scale and the high number of points over the

whole research area, the visualization of the spatial analyses on continental scale is challenging. Therefore, the research area is split up through an overlay of a 2.5° grid (as shown in fig. 4). The overall and class-wise accuracy results for all points within each 2.5° grid cell are aggregated in order to identify large scale spatial quality differences for the analyzed LULC groups. Additionally, the total number of points for each LULC group per grid cell are displayed in section 4.3.

### 4.2  Accuracy assessment - background & design

The key components of the accuracy assessment of a large-scale land cover product are **objective**, **sampling design**, **response design** and the final **analyses and estimation** (Wulder et al., 2006). All of the key components have great impact on the quality of the assessment and further, on the final metrics, especially in the present assessment, where reference and assessed dataset differ widely in structure. LANDMATE PFT is a gridded dataset with fractional LULC classes but no information on the subgrid location within the grid cell. Other than that, the points of GT-SUR have fixed locations expressed through

exact coordinates, but no (exact) information on the spatial extent of this class. Another challenge is the fractional structure of LANDMATE PFT itself, where one unit (grid cell) possibly contains multiple fractions. Therefore, the design of the accuracy assessment needs to be customized to the **objective**, which is to determine the overall quality of the LANDMATE PFT map for Europe 2015 as well as the quality of individual LULC type representation within the map in order to derive recommendations for the use of LANDMATE PFTs in RCMs.

When it comes to the **sampling design**, sampling size, spatial distribution of the respective sample and the representation of each LULC group or class within the sample are crucial to produce reliable quality information about a LULC product (Stehman, 2009). However, the collection of ground truth data is a rather expensive procedure regarding time and money, which needs to be considered during the process. The sample size is therefore a compromise size and cost. In the present assessment, an existing ground truth database containing over 340,000 records is used as reference which eliminates the possible issue of

a too small sample size. It is also known that all assessed LULC groups are represented in a sufficiently high number (Table 6). Nevertheless, the present assessment is a special case situation with every unit of LANDMATE PFT containing more than one LULC group potentially. Therefore, the subsets are selected through application of a filter to capture the map accuracy in a way that accounts for the fractional structure within the grid cells in the LANDMATE PFT map (see section 4.2.1).

The **response design** deals with the spatial support regions (SSR) and the labelling protocol or classification harmonization.

The SSR is a buffer region around a sampling unit that is selected to account for small-scale landscape heterogeneity that is likely not captured by larger scale map products. In the present case, the sampling design is selected in a way that the grid cells of LANDMATE PFT serve as SSR for each GT-SUR point. A fraction is not located precisely at one location within the respective grid cell but evenly distributed over the whole grid cell. Assuming, the uniformly distributed fraction can occur



in small patches or in one large patch within the grid cell, the whole grid cell is defined as SSR for the respective LULC
group. The labelling protocol needs to be determined to deal with the different legends of the reference and the assessed map.
The harmonization of legends is selected in regard to the objective of the respective assessment, as in this case, to provide
information about the quality of representation of the most dominant LULC types in LANDMATE PFT. The labelling protocol
used in the present assessment is summarized in table 5.

The **analyses and estimation** used are error matrices, that give an overview of the overall and LULC group-wise accuracy of
the LANDMATE PFT map. For both resolutions of LANDMATE PFT, the error matrices and the resulting accuracy measures
overall accuracy (OA), producer's accuracy (PA) and user's accuracy (UA) are calculated, where PA and OA are calculated
group-wise. The error matrix is a cross-tabulation between map and reference of the size $q$ x $q$, where $q$ stands for the number
of land cover classes or groups. The map classes are placed in the rows and the reference classes in the columns so that
the diagonal of the matrix gives the sum of the correctly classified map units. The off-diagonal cell values represent the
disagreement between the map and the reference. The overall accuracy is calculated according to equation 1:

$$OA = \frac{\sum_{i=1}^{q} n_{ii}}{n} * 100 \tag{1}$$

The sum of the agreeing diagonal elements $n_{ii}$ of all LULC groups is divided by the number of all observations $n$. The
PA represents the accuracy from the view of the map producer. The PA stands for the probability, that a LULC feature in the
reference is classified as the respective feature by the map. The PA is calculated using equation 2 where the number of correctly
classified units per LULC group $n_{ii}$ is divided by the total number of LULC group occurrences of the reference $n_{+i}$:

$$PA_i = \frac{n_{ii}}{n_{+i}} * 100 \tag{2}$$

While the PA gives the proportion of features in the reference that are actually represented as those in the produced map,
the UA is the accuracy from the perspective of the map user. It is the probability of a feature classified as such in the map is
actually present in the reference. The UA is calculated using equation 3, where the number of correctly classified pixels $n_{ii}$ per
LULC group is divided by the row sum $n_{i+}$ $\sum_{i=1}^{p} n_{ji}$:

$$UA_i = \frac{n_{ii}}{n_{i+}} * 100 \tag{3}$$

### 4.2.1 Dataset harmonization & filter

The quality assessment is done assigning the PFT type with the maximum fraction per grid cell to the GT-SUR points located
within respective grid cell. The classifications of both datasets need to be harmonized in order to make the comparison as
detailed as possible but also to be able to produce reliable and robust results for the RCM community. For the analysis, the
classifications of LANDMATE PFT and the GT-SUR are harmonized as shown in table 5.





**Table 5.** Classification harmonization between LANDMATE PFT map and GT-SUR

| GT-SUR LC group | GT-SUR group name | LANDMATE PFT number | LANDMATE PFT name | Harmonization group number | Harmonization name |
|---|---|---|---|---|---|
| A | Artificial Land | 15 | Urban | 1 | URBAN |
| B | Cropland | 13 | Non-irrigated Crops | 2 | CROPLAND |
| | | 14 | Irrigatred crops | | |
| C | Woodland | 1 | Tropical broadleaf evergreen trees | 3 | WOODLAND |
| | | 2 | Tropical deciduous trees | | |
| | | 3 | Temperate broadleaf evergreen trees | | |
| | | 4 | Temperate deciduous trees | | |
| | | 5 | Evergreen coniferous trees | | |
| | | 6 | Evergreen deciduous trees | | |
| D | Shrubland | 7 | Coniferous shrubs | 4 | SHRUBLAND |
| | | 8 | Deciduous shrubs | | |
| E | Grassland | 9 | C3 Grass | 5 | GRASSLAND |
| | | 10 | C4 Grass | | |
| F | Bare land | 16 | Bare | 6 | BARE AREAS |
| G | Water | 11 | Tundra | 7 | OTHER |
| H | Wetlands | 12 | Swamps | | |
| Other | Marine areas | | | | |

The LULC groups URBAN, CROPLAND, WOODLAND, SHRUBLAND, GRASSLAND, and BARE ARES are harmonized without applying modifications to the classifications. The LANDMATE PFTs can easily be grouped or directly adopted while the GT-SUR level one classification (letters A-H) is completely adopted into the harmonized groups. The LANDMATE PFT map is a product developed for the use in RCMs. In general, RCMs implement a land-sea-mask to determine aquatic areas for both, inland and marine water. Therefore, the categories WATER and MARINE areas are neglected in the analyses. The LANDMATE PFTs "Tundra" and "Swamp" can not be assigned with a sufficient agreement to the GT-SUR class definitions. Therefore, the GT-SUR groups water, wetlands and marine areas as well as the LANDMATE PFTs Tundra and Swamps are merged into the group "OTHER" for the assessment. Although the group cannot be evaluated regarding the quality of the

330 LANDMATE PFT map, the group needs to be involved in the assessment to keep the numbers in the assessment correct and reliable for all other groups.

Both datasets are provided in a regular Gaussian grid (WGS84 EPSG:4326) so that no reprojection of the datasets needs to be done for the comparison. The descriptive statistics for each LULC group for the reference GT-SUR and the LANDMATE PFT dataset are summarized in table 6.

**Table 6.** General information on data in the comparison

| LULC group[11] | GT-SUR[12] | LANDMATE PFT 0.018°[13] | Dominant LANDMATE PFT 0.018°[14] |
|---|---|---|---|
| URBAN | 14,393 | 65,000 | 7,577 |
| CROPLAND | 83,295 | 248,301 | 136,970 |
| WOODLAND | 124,374 | 277,290 | 124,437 |
| SHRUBLAND | 27,298 | 302,035 | 19,790 |
| GRASSLAND | 66,541 | 333,948 | 44,244 |
| BARE AREAS | 10,395 | 31,756 | 4,148 |
| OTHER | 12,340 | 28,823 | 1,470 |
| Sums | 338,636 | | 338,636 |

The LANDMATE PFT dataset includes multiple LULC fractions per grid cell. Accordingly, the area proportion of the dominant LULC group varies widely and thus the likelihood that the GT-SUR point sample falls within this area. The filter applied is categorizing the grid cells regarding the proportion of the dominant LULC group, the higher the threshold, the stricter the filter and the more likely a specific sample falls into the subgrid fraction of the dominant class. The filter set numbers 1-10 are representing the cells containing a minimum of 10 - 100% of the dominant LULC group according to table 7.

---

[11] LULC group analyzed in the quality assessment

[12] GT-SUR points assigned per LULC group

[13] number of grid cells in LANDMATE PFT that have a share of the respective LULC group >0%

[14] Sum of LANMDATE PFT grid cells where the respective LULC group is represented dominantly



**Table 7.** Filter sets with varying dominant LULC group share per grid cell from >10% to 100%

| Filter set number | fraction size of dominant LULC group | LANDMATE PFT cells within filter set |
|---|---|---|
| 1 | >10 % | 338619 |
| 2 | >20 % | 338619 |
| 3 | >30 % | 336703 |
| 4 | >40 % | 311238 |
| 5 | >50 % | 259073 |
| 6 | >60 % | 203343 |
| 7 | >70 % | 137412 |
| 8 | >80 % | 74765 |
| 9 | >90 % | 26993 |
| 10 | 100 % | 1449 |

## 4.3 Results

In order to show the impact of the applied spatial filter, the spatial distribution of agreement and disagreement of LANDMATE PFT with the reference GT-SUR is investigated. The point counts and percentage agreement are aggregated and averaged, respectively, per 2.5° grid cell. After giving an overview over the overall accuracy measures the individual LULC group results are discussed in the following subsections. Note that due to the low overall point count as shown in table 6, the LULC groups

SHRUBLAND and BARE AREAS are discussed together in section 4.3.5. Figure 5 shows the spatial distribution of the filter sets over Europe while fig. 6 shows the overall accuracy for the filter sets.

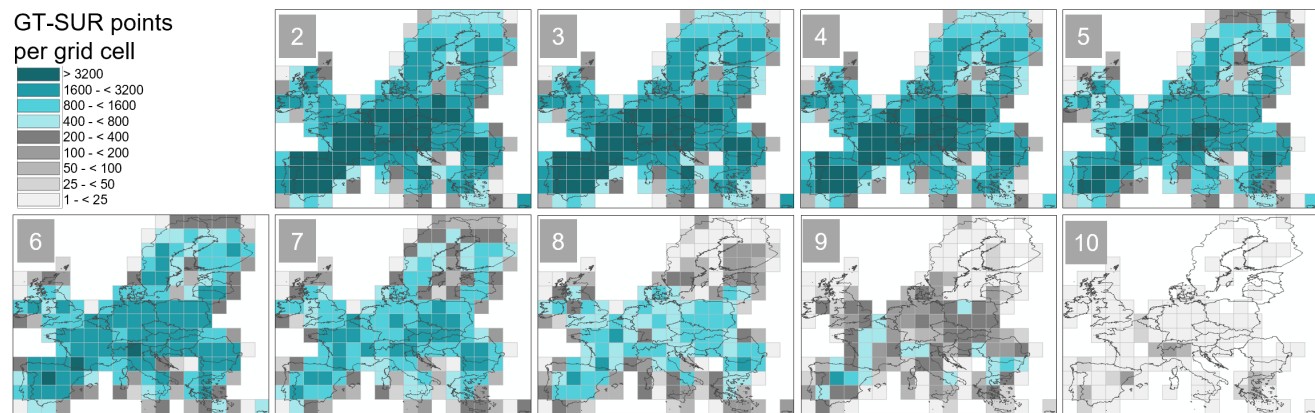

**Figure 5.** The distribution of the varying dominant LULC group filter sets over the research area in Europe. Since the >10% and the >20% filter set share the same number of points the >10% filter set is not shown.





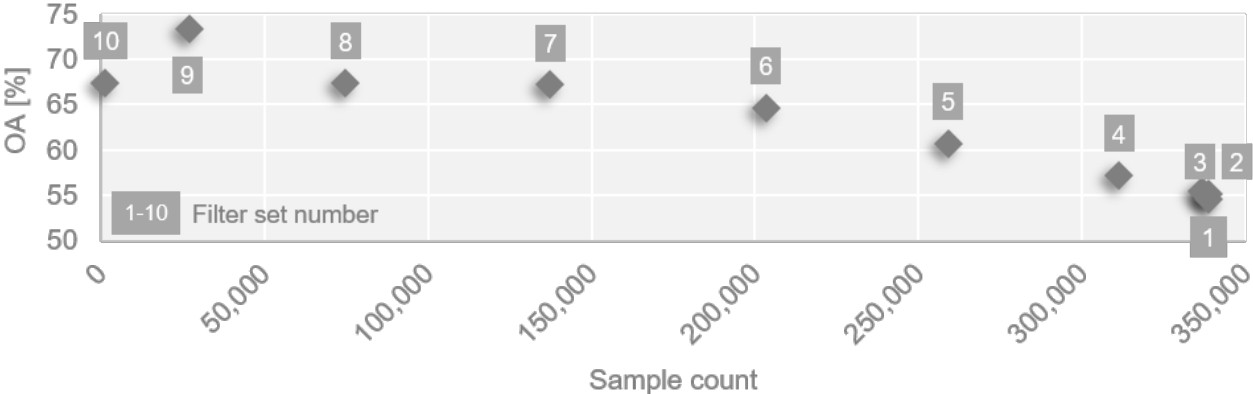

**Figure 6.** Overall accuracy for the full domain of the 10 filter sets as introduced in table 7 as function of filter set size. Filter set numbers are shown in grey boxes.

In order to be able to capture the LULC group diversity and distribution characteristics of LANDMATE PFT, the filter set must be distributed well over the respective area and contain a sufficiently large proportion of the total cells. Figure 5 shows that the filter sets are distributed reasonably well up to filter set 7. Filter set 8 shows a quite patchy pattern and a strongly

decreasing sample number in Northern Europe. Within filter set 9, the patchy pattern of low sample count per 2.5° grid cell spreads over the whole continent. While filter set 9 could still be used for evaluation of LANDMATE PFT for limited regions in Europe, filter set 10 is clearly not evaluable due to the overall small sample count (< 1500). This pattern is also found for filter sets 9 & 10 of the individual LULC groups. The filter set sizes as well as the applied filter itself have direct impact on the OA shown in fig. 6. The decreasing OA towards the higher sample count is an effect of the LANDMATE PFT grid cell

heterogeneity representation in each filter set. Filter set 1-3 include all LANDMATE PFT grid cells where the dominant LULC group occupies a minimum of 10% to 30% respectively. Therefore, the probability that the GT-SUR point sample within the respective grid cell represents a location that is occupied by one of the non-dominant LULC groups is relatively high. The applied filter accounts for the impact of the structure difference of the two datasets. The higher probability of agreement is reflected in the increasing OA for the samples that include only grid cells with an occupation of 50% or more of the dominant

LULC group. Sample 9 & 10 represent the LANDMATE PFT dataset not adequately regarding distribution and diversity while sample 8 shows a poor coverage in northern Europe. In order to include the largest proportion of the total sample in the analysis, the point count per LULC group as well as the PA per 2.5° grid cell for filter set 2 is analyzed in the results section (fig. 7). In order to give an overview of the spatial accuracy for the evaluable filter range, The respective figures for filter set 5 and 7 are shown in Appendix B (tables B1 & B2).

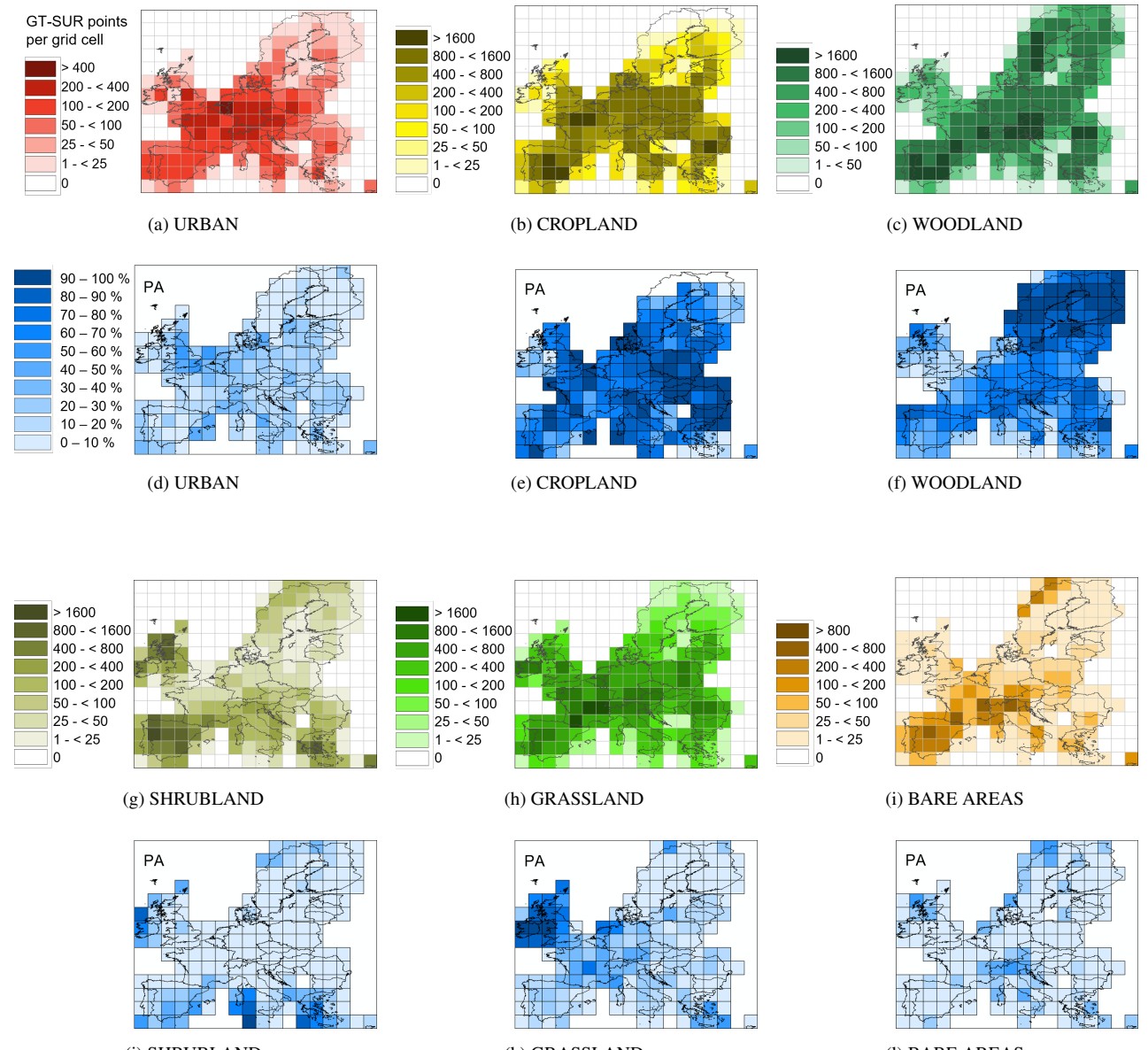

**Figure 7.** Total count of GT-SUR points per 2.5° grid cell (a-c; g-i) and producer's accuracy for the individual LULC groups (d-f;j-l) for filter set 2 (dominant LULC group occupies > 20% per LANDMATE PFT grid cell)

### 4.3.1 URBAN

The urban representation in LANDMATE PFT for filter set 2 is shown in fig. 7a and 7d. The PA for the filter sets 1-10 is shown in fig. 8 where an overall low PA for all filter sets is found. With increasing proportion of the dominant LULC group URBAN





the PA increases slightly but is still lower than 40% for samples that include enough points to be considered representative for the research area. The overall low PA is reflected in the URBAN maps in fig. 7 as well as in fig. B1 and B2.

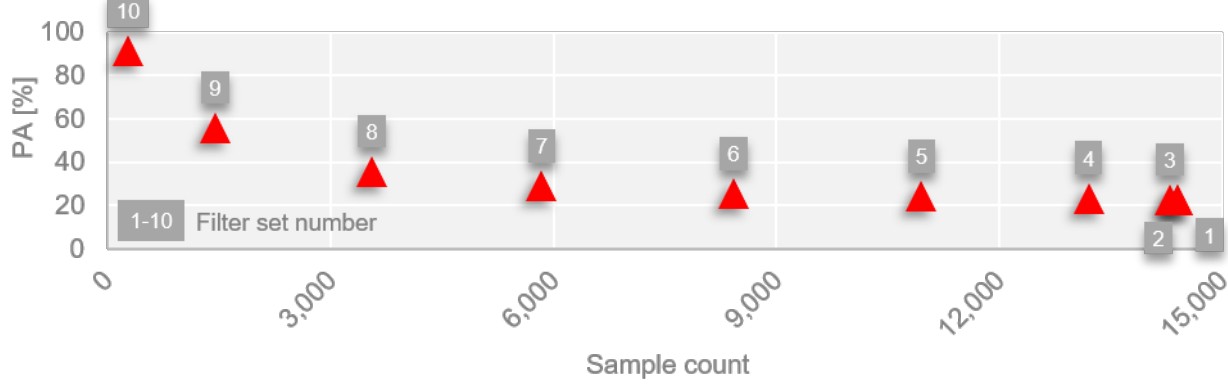

**Figure 8.** Producer's accuracy of the 10 filter sets (Filter set numbers in grey boxes) for the LULC group URBAN as a function of sample count per filter set.

A visual check of the map agreement between LANDAMTE PFT and GT-SUR revealed the issue that leads to the overall low PA. Figure 9 shows four large URBAN agglomerations in different areas of Europe where the red points represent GT-SUR urban points while the white points represent GT-SUR point representing non-urban LULC groups. The grey-scaled squares represent the LANDMATE PFT URBAN fractions from zero (no coverage, white) to one (full coverage, black) within one grid cell.

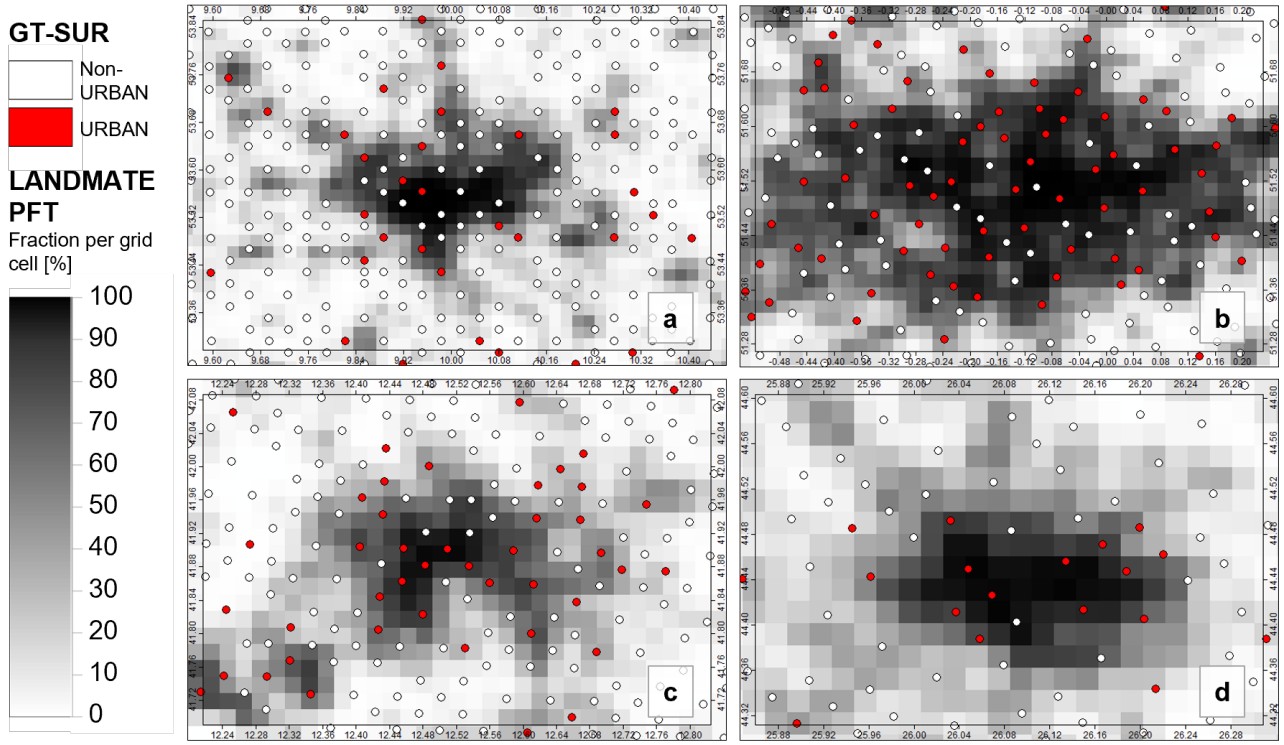

**Figure 9.** Examples of URBAN representation in LANDMATE PFT (greyscale grid) and GT-SUR (points). Cities shown are Hamburg (a), London (b), Rome (c) and Bukarest (d).

The LANDMATE PFT grid cells with a large urban fraction indicate the respective city core while the GT-SUR points that are located within the city core are mostly not classified as URBAN. However, the GT-SUR points do not fail to represent the structure of urban areas because they are characterized through a heterogeneous pattern of sealed surfaces, recreational areas (e.g. parks) and different building types and density, not through a homogeneous sealed area. The LANDMATE PFT map represents this heterogeneous structure through the varying fractions of non-urban PFTs within the grid cell. However, in order to make the impact of a larger city visible in an RCM simulation, it is beneficial for LANDMATE PFT to represent a larger city with a dense core structure. In order to verify the representation of the large URBAN agglomerations in Europe, a comparison with the World Settlement Footprint for 2015 (WSF, Marconcini et al., 2020) dataset was done (not shown). The comparison showed that not only larger agglomerations but also smaller patches of settlements are represented well in LANDMATE PFT. Therefore, despite the low agreement with GT-SUR in the present assessment, the URBAN PFT of LANDMATE PFT 2015 is of sufficiently good quality and suitable to represent urban land cover in high resolution (~2 km) RCM simulations. Due to the abovementioned comparability issues the UA of the LULC group URBAN will not be further discussed.





### 4.3.2 CROPLAND

The CROPLAND representation in LANDMATE PFT shows, together with WOODLAND the highest PA for the research area. As shown in fig. 10 the PA for all filter sets is > 80% which is to be considered as a very good agreement with the reference.

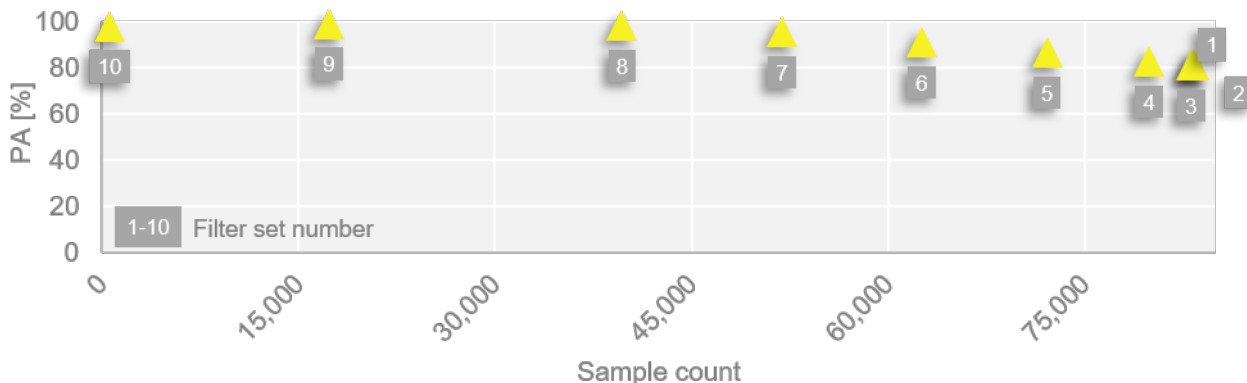

**Figure 10.** Producer's accuracy of the 10 filter sets (Filter set numbers in grey boxes) for the LULC group CROPLAND as a function of sample count per filter set.

Figure 7b shows the distribution of CROPLAND points in GT-SUR over the research area. CROPLAND points are the second most frequent LULC group in GT-SUR and are mainly distributed over middle and southern Europe. Although the northern European grid cells show a lower count of CROPLAND points, figure 7e shows that the PA is still very high in these areas. The PA increases with increasing filter set homogeneity (Fig. B1 and B2). Regarding the UA for CROPLAND, LANDMATE PFT shows a strong overestimation, where ~51% of the LANDMATE PFT CROPLAND cells in filter set 2 are actually another LULC group in the reference. More than half of the LANDMATE PFT CROPLAND areas are mostly WOODLAND, GRASSLAND, and a mix of the other LULC groups in the reference. The UA for CROPLAND increases rapidly towards the more homogeneous filter sets (~61% for filter set 7). However, the confusion with WOODLAND and GRASSLAND is non-negligible and will be discussed in section 6.

### 4.3.3 WOODLAND

For the representation of WOODLAND, the PA shows the second highest values with > 70% for all filter sets with a reasonably high point count (filter sets 1-7, fig. 11). Similar to CROPLAND, the sampling filter does not have a large impact on PA. The highest PA is reached over the northern European regions (Fig. 7f). Deficits are visible over the southern British Isles, some parts of France and the coastline along Belgium and the Netherlands. Further, the Mediterranean Coast shows a low PA within grid cells that have an overall small point count (Fig. 7c).



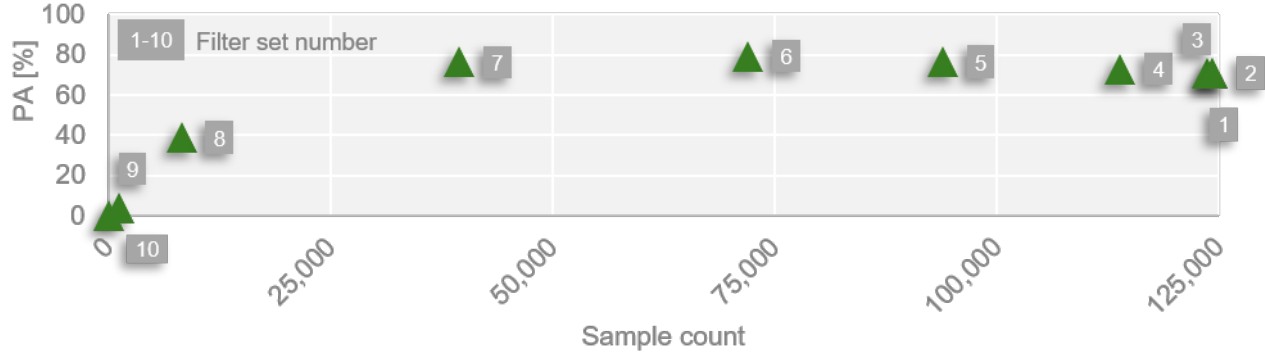

**Figure 11.** Producer's accuracy of the 10 filter sets (Filter set numbers in grey boxes) for the LULC group WOODLAND as a function of sample count per filter set.

The differences between northern and southern regions tends to increase towards the more homogeneous filter sets as shown in figures B1f and B2f. Agreement over the northern regions increases while agreement over the Iberian Peninsula decreases together with a rapid decrease of the filter set count within the corresponding grid cells. The UA for WOODLAND is noticeably higher than for all other LULC groups (> 70% for filter set 2 and increasing towards the more homogeneous filter sets) which emphasises the very good quality of WOODLAND representation in LANDMATE PFT. (~10% for filter set 2). Further, ~4% of the total LANDMATE PFT cells representing WOODLAND are actually CROPLAND or OTHER.

### 4.3.4 GRASSLAND

The GT-SUR sampling points show the highest GRASSLAND coverage in central Europe with the highest occurrence in Ireland and the southern part of France (Fig. 7h). The PA for LANDMATE PFT GRASSLAND according to fig. 7k is not noticeably higher in these areas but overall highest in the Southwest of the British Isles. For all filter sets, the PA ranges between 32 and 34% which is considerably low (Fig. 12.



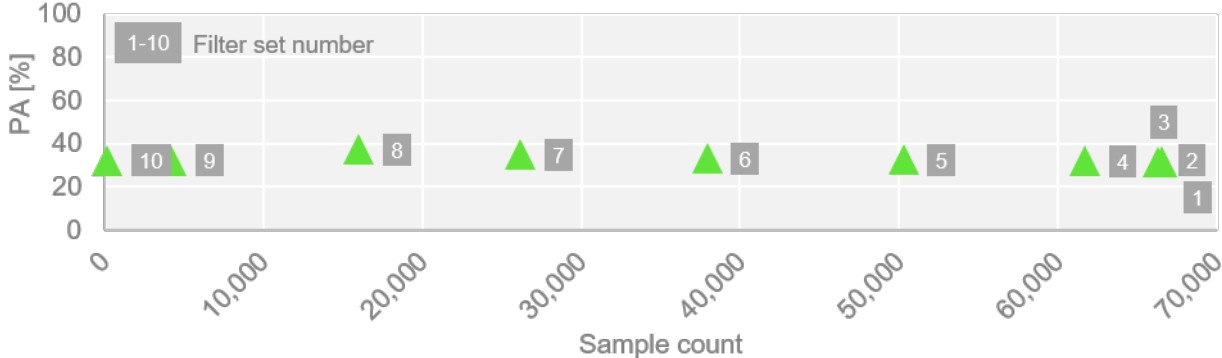

**Figure 12.** Producer's accuracy of the 10 filter sets (Filter set numbers in grey boxes) for the LULC group GRASSLAND as a function of sample count per filter set.

One reason for this low accuracy of LANDMATE PFT regarding GRASSLAND can be found looking at the results of sections 4.3.2 and 4.3.3. The UAs of CROPLAND and WOODLAND reveal that ~20% of the LANDMATE PFT CROPLAND cells and ~10% of the LANDMATE PFT WOODLAND cells are actually representing GRASSLAND in the reference, which
adds up to over 60% of the total GT-SUR GRASSLAND points. Another reason is found in the dataset structure of LANDMATE PFT. A considerable amount of GRASSLAND is not part of the assessment because GRASSLAND does not make the dominant but the second dominant PFT in many grid cells (~45% of all LANDMATE PFT grid cells). Therefore,the seemingly weak GRASSLAND representation in LANDMATE PFT rather shows a weakness of the present assessment that is caused by the different dataset structures.

**4.3.5 SHRUBLAND & BARE AREAS**

The PA for SHRUBLAND and BARE AREAS is the lowest of all assessed LULC groups with < 20% for all filter sets of both LULC groups respectively (Fig. 13 and 14). The low point count of both LULC groups might be one reason for the low PA. However, looking at the distribution of the SHRUBLAND and BARE AREA points in fig. 7i, LANDMATE PFT is not able to capture the LULC groups even in grid cells with a relatively high point count. The GT-SUR shows ~27,000 SHRUBLAND
points while LANDMATE PFT shows only ~19,000. Therefore, one reason for the poor SHRUBLAND representation lies within the base map (ESA-CCI LC) used for the creation of LANDMATE PFT, where the known small count of SHRUBLAND proportions was inherited by LANDMATE PFT. It must be noted, that a large proportion of SHRUBLAND in ESA-CCI LC is part of the mixed LC classes, such as Shrubland/Cropland or Shrubland/Forest. The known deficit was partly compensated by the translation into the PFTs, where SHRUBLAND proportions were added to the total as proportions of the mixed ESA-CCI
LC classes. Further SHRUBLAND makes the second dominant PFT in ~20% of the total LANDMATE PFT grid cells in the assessment. Just like for GRASSLAND, these SHRUBLAND proportions can not be addressed within the present assessment.



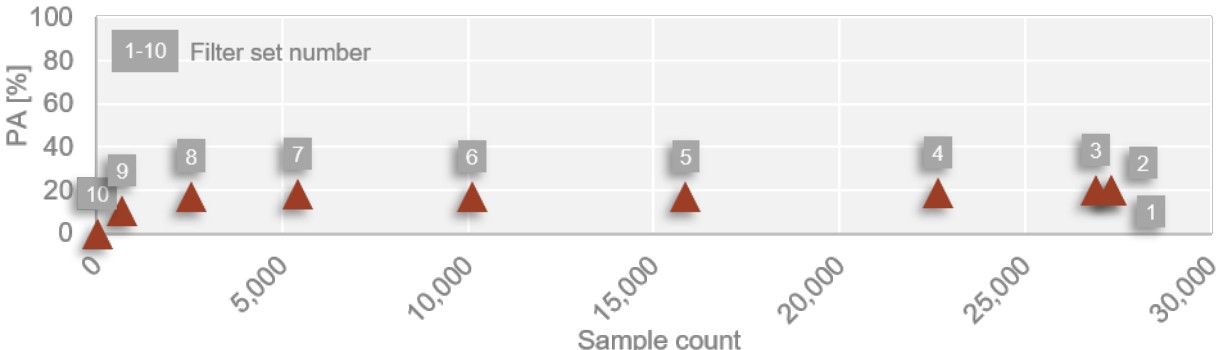

**Figure 13.** Producer's accuracy of the 10 filter sets (Filter set numbers in grey boxes) for the LULC group SHRUBLAND as a function of sample count per filter set.

The overall BARE AREAS sample count in LANDMATE PFT in filter set 2 is < 50% of the actual BARE AREA points in GT-SUR. Almost half of the GT-SUR BARE AREAS points are identified as CROPLAND while ~30% are identified as WOODLAND or GRASSLAND. Only ~17% (< 2,000 points for filter set 2) of the GT-SUR BARE AREAS are actually
identified by LANDMATE PFT with the largest agreement in the Alps, Northern Great Britain, and Northern Scandinavia (Fig. 7l. However, due to the comparably low sample count the spatial assessment is not robust. Just like for SHRUBLAND, the homogeneity of LANDMATE PFT cells does not have a large impact on the PA. UA is higher than PA with ~43% and increasing towards the more homogeneous filter sets. However, considering the rapidly decreasing sample count for the more homogeneous filter sets, the accuracy measures are becoming even less representative for the BARE AREA representation in
LANDMATE PFT. Nevertheless, the confusion with the other LULC groups is further discussed in section 6.

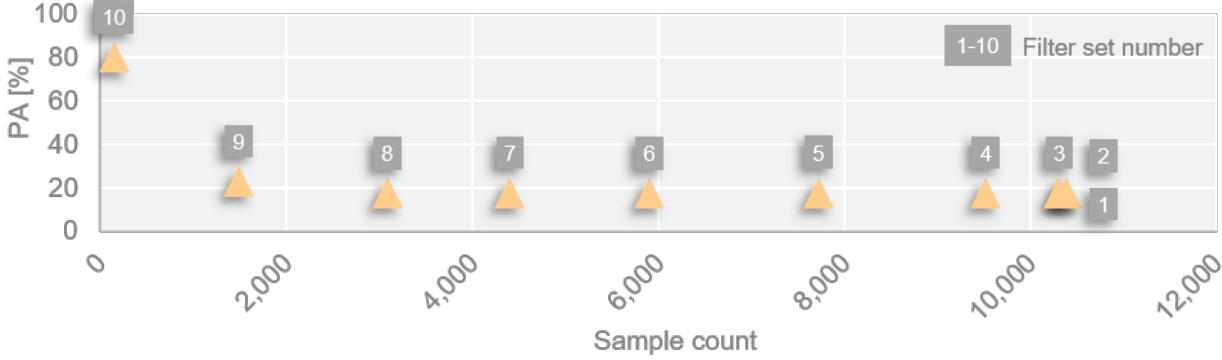

**Figure 14.** Producer's accuracy of the 10 filter sets (Filter set numbers in grey boxes) for the LULC group BARE AREAS as a function of sample count per filter set.



## 5 Data availability

The LANDMATE PFT dataset for Europe 2015 is published with the Long Term Archiving Service (LTA) for large research datasets, which are relevant for climate or earth system research, of the German Climate Computing Service (DKRZ). As World Data Center for Climate (WDCC), the DKRZ LTA is accredited as regular member of the World Data System. The

450 LANDMATE PFT dataset for Europe 2015 is available within the LANDMATE project data at https://cera-www.dkrz.de/WDCC/ui/cerasearch/entry?acronym=LM_PFT_LandCov_EUR2015_v1.0 (Reinhart et al., 2021b). Within the LANDMATE project, a short documentation summarizes the technical information corresponding to LANDMATE PFT.

## 6 Discussion & conclusion

The present work introduces the preparation of the LANDMATE PFT map for the European Continent based on several LULC

datasets and climate data.

The LANDMATE PFT Version 1.0 is prepared in order to provide realistic, high-resolution LULC representation for RCMs. The dataset includes LULC information from different, validated sources as well as regional climate information through involvement of the HLZs. For each ESA-CCI land cover class, an individual CWT is developed to translate the original LULC classes into PFTs. The various mixed LULC classes included in the base map ESA-CCI LC are extremely difficult to

460 resolve within RCMs. Through the developed CWP, the mixed LULC classes can be disaggregated into PFT fractions, which improves the realistic representation of these classes in RCMs. The involvement of the climate data further allows a customized translation of LULC classes for individual regions. The 16 LANDMATE PFTs are selected to provide simple transferability into various RCM families in order to be able to conduct coordinated RCM experiments where the implementation of a common, high quality LULC map provides minimum uncertainty for a multi-model ensemble.

The accuracy assessment of LANDMATE PFT is conducted in the form of a comparison with the ground truth dataset GT-SUR. In order to account for the different structure of the reference GT-SUR and the assessed LANDMATE PFT map and further the fractional structure of the LANDMATE PFT grid cells, a filter is applied. All filtered LANDMATE PFT subsets are analyzed in terms of agreement with the reference (i.e., GT-SUR). In order to investigate regional differences in accuracy measures, a spatial analysis supported by gridded maps of the research area is done. The quality of the LANDMATE PFT

map is assessed using the overall accuracy (OA) and the producer's and user's accuracy (PA and UA) for the individual LULC groups. Overall, the assessment serves as recommendation and uncertainty information for regional climate modellers that use LANDMATE PFT, or the time series LUCAS LUC (Hoffmann et al., submitted), which is based on LANDMATE PFT, in RCMs.

Within the accuracy assessment, the OA does not change considerably between the evaluable filter sets of the respective

LULC groups which shows that the dataset structure has no noticeable impact on that accuracy measure. The highest PA is found for CROPLAND and WOODLAND which are the dominant LULC groups in the research area. The lowest PA is found for SHRUBLAND and BARE AREAS, which are also the LULC groups with the lowest overall sample count. The UA is found to be highest for WOODLAND, followed by CROPLAND, GRASSLAND and BARE AREAS. Both accuracy measures, PA





and UA are highly influenced by the proportion of the dominant LULC group in the individual grid cell. The difference between the filter sets for UA of the LULC groups is 10 to 20% per group while the difference for PA is noticeable but considerably lower, which means that the applied filter has a higher influence on the former.

The URBAN representation in LANDMATE PFT represents a special case in the present assessment due to the heterogeneous structure of urban areas. Both datasets, GT-SUR and LANDMATE PFT are able to represent the LULC group URBAN very well for their respective purpose. Nevertheless, the PA for URBAN reflects the limitations of the present assessment method. The fine scale point data of GT-SUR represents the patchwork structure of recreational areas, building blocks, and other urban elements at the location of the respective points while LANDMATE PFT represents the urban area as an agglomeration of grid cells with URBAN as the dominant LULC group. The additional comparison with a high resolution dataset (WSF2015) showed that not only large but also small agglomerations of urban areas are represented well in LANDMATE PFT. Therefore and despite of the accuracy assessment results for the LULC group URBAN, the LANDMATE PFT dataset can be recommended to be used in RCMs that resolve urban features over the European Continent.

A limitation of LANDMATE PFT is the overestimation of CROPLAND to the expense of WOODLAND and GRASSLAND and the overestimation of WOODLAND to the expense of mostly GRASSLAND. This overestimation has a minor impact on the overall WOODLAND and CROPLAND representation but a major impact on the representation of GRASSLAND in LANDMATE PFT. The representation of GRASSLAND is comparably low due to the aforementioned reasons. Further, the LULC groups with the lowest point counts SHRUBLAND and BARE AREAS are not well represented, which happens due to the low overall sample size but also due to the overall too low representation in LANDMATE PFT, which is partly inherited from the base map ESA-CCI LC. The representation of these LULC groups needs to be considered when using LANDMATE PFT in RCM simulations using the supporting maps in fig. 7,B1 and B2.

The representation of LULC groups in LANDMATE PFT is assessed through the comparison with ground truth data. The structural differences of the datasets, where gridded data is compared to point data, is a major weakness of this assessment. Although the fractional structure does not have a major influence on the OA, the LULC group-wise PA and even more the UA is affected.

The present assessment takes into account the dominant LULC group per grid cell of LANDMATE PFT. Depending on the proportion of this LULC group, the second or third-most represented LULC group can occupy a considerable area of the respective grid cell. Therefore, a follow up assessment, where these LULC group proportions are also considered and compared to the ground truth is needed in order to investigate, if the PA of the less dominant LULC groups GRASSLAND, SHRUBLAND, and BARE AREAS is increased. The use of additional LULC data, like it was done for URBAN in this assessment, would be an additional useful step to validate the quality of GRASSLAND, SHRUBLAND and BARE AREAS representation in LANDMATE PFT.

The results show that the LANDMATE PFT map is able to represent LULC over large parts of Europe in a sufficient quality. Especially the dominant LULC groups are represented overall well which is highly beneficial for RCM experiments that require realistic, high-resolution LULC representation. Nevertheless, there are uncertainties found for the less represented LULC groups. When using LANDMATE PFT in an RCM it is crucial to consider these uncertainties when interpreting simu-





lation results. Especially the spatial distribution of uncertainties in LANDMATE PFT needs to be considered when comparing

simulation results to observations because the input parameters in the employed land-surface schemes are influenced by the individual LULC, which subsequently considerably impacts on lower-atmosphere processes, such as the intensity of heat and moisture exchange. Thus, by carefully considering the issue of uncertainty introduced by the LULC input, misconclusions about RCM model performance and about small-scale interconnections can be avoided (Ge et al., 2007; Sertel et al., 2010; Santos-Alamillos et al., 2015; Reinhart et al., 2021a).

Beside the quality of the LULC product, the implementation process of each individual RCM is crucial for the realistic representation of LULC in regional climate model experiments. When translating a LULC product into the model specific LULC classes and structure, modifications are done that can change the map characteristics. When the LANDMATE PFT product is used in an RCM that only uses the dominant LULC fraction per grid cell, the overall LULC proportions can change. The same applies when LANDMATE PFT is used in a model with limited fractions per grid cell or a different classification

system. The present assessment gives a guideline on the quality of LANDMATE PFT (Version 1.0) when used unaltered. Through the involvement of the ground truth data, regional deficits of LANDMATE PFT are presented that can be compensated during the implementation process into the individual RCM or RCM family.

The findings of the present assessment support the identification of uncertainties within the LANDMATE PFT map for Europe. Nevertheless, user feedback is crucial for the future overall improvement of LANDMATE PFT. The RCM community

within the WCRP FPS LUCAS is already participating in the feedback process where implementation of LANDMATE PFT and the LUCAS LUC time series into different RCMs is comprehensively documented. The future work on LANDMATE PFT also includes the extension of the dataset to other CORDEX regions. Although, the dataset is based on various globally available datasets and therefore, can be created globally, the introduced quality assessment method must be performed for each region individually, desirably using region-specific expert knowledge. Further, the assessment should be expanded in order to

include the second or third-most represented LULC group per grid cell to possibly achieve more accurate quality information about LANDMATE PFT.



# Appendix A

**Table A1.** Cross-walking table for ESA-CCI LC class 10 - Cropland, rainfed and LC class 11 -Cropland, herbaceous cover. For LC class 10 and 11, no HLZ were assigned

| | 1 | 2 | 3 | 4 | 5 | 6 | 7 | 8 | 9 | 10 | 11 | 12 | 13 | 14 | 15 | 16 |
|---|---|---|---|---|---|---|---|---|---|---|---|---|---|---|---|---|
| | Tree | | | | | | Shrub | | Grass | | Special vegetation | | Crops | | Non-vegetated | |
| **Holdridge Life Zone** | tropical broadleaf evergreen | tropical broadleaf deciduous | temperate broadleaf evergreen | temperate broadleaf deciduous | evergreen coniferous | deciduous coniferous | evergreen | deciduous | C3 | C4 | Tundra | Swamps | crops | | urban | bare ground |
| 1-30 | | | | | | | | | 10 | | | | 90 | | | |

**Table A2.** Cross-walking table for ESA-CCI LC class 12 - Cropland, tree or shrub cover. For LC class 12, no HLZ were assigned

| | 1 | 2 | 3 | 4 | 5 | 6 | 7 | 8 | 9 | 10 | 11 | 12 | 13 | 14 | 15 | 16 |
|---|---|---|---|---|---|---|---|---|---|---|---|---|---|---|---|---|
| | | | | | | | Shrub | | Grass | | Special vegetation | | Crops | | Non-vegetated | |
| **Holdridge Life Zone** | tropical broadleaf evergreen | tropical broadleaf deciduous | temperate broadleaf evergreen | temperate broadleaf deciduous | evergreen coniferous | deciduous coniferous | evergreen | deciduous | C3 | C4 | Tundra | Swamps | crops | | urban | bare ground |
| 1-30 | | | | | | | | 70 | | | | | 30 | | | |

**Table A3.** Cross-walking table for ESA-CCI LC class 20 - Cropland, irrigated or post flooding. For LC class 20, no HLZ were assigned

| | 1 | 2 | 3 | 4 | 5 | 6 | 7 | 8 | 9 | 10 | 11 | 12 | 13 | 14 | 15 | 16 |
|---|---|---|---|---|---|---|---|---|---|---|---|---|---|---|---|---|
| | Tree | | | | | | Shrub | | Grass | | Special vegetation | | Crops | | Non-vegetated | |
| **Holdridge Life Zone** | tropical broadleaf evergreen | tropical broadleaf deciduous | temperate broadleaf evergreen | temperate broadleaf deciduous | evergreen coniferous | deciduous coniferous | evergreen | deciduous | C3 | C4 | Tundra | Swamps | crops | | urban | bare ground |
| 1-30 | | | | | | | | | | | | | 100 | | | |



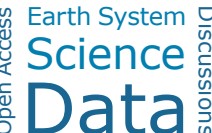

**Table A4.** Cross-walking table for ESA-CCI LC class 30 - Mosaic cropland (>50%) / natural vegetation (tree, shrub, herbaceous cover)(<50%).

| | 1 | 2 | 3 | 4 | 5 | 6 | 7 | 8 | 9 | 10 | 11 | 12 | 13 | 14 | 15 | 16 |
|---|---|---|---|---|---|---|---|---|---|---|---|---|---|---|---|---|
| | Tree | | | | | | Shrub | | Grass | | Special vegetation | | Crops | | Non-vegetated | |
| Holdridge Life Zone | tropical broadleaf evergreen | tropical broadleaf deciduous | temperate broadleaf evergreen | temperate broadleaf deciduous | evergreen coniferous | deciduous coniferous | evergreen | deciduous | C3 | C4 | Tundra | Swamps | crops | | urban | bare ground |
| 1-6 | | | | | | | | | | | 20 | 20 | 60 | | | |
| 7-9 | | | | | | | | | 40 | | | | 60 | | | |
| 10 | | | | 10 | | | | | 30 | | | | 60 | | | |
| 11,12 | | | | 30 | | | | | 10 | | | | 60 | | | |
| 13,14 | | | | | | | | | 40 | | | | 60 | | | |
| 15 | | | | 5 | 5 | | | 20 | 10 | | | | 60 | | | |
| 16 | | | | 7.5 | 7.5 | | | 10 | 15 | | | | 60 | | | |
| 17,18 | | | | 20 | | | | 10 | 10 | | | | 60 | | | |
| 19 | | | | | | | | | 40 | | | | 60 | | | |
| 20 | | | | | | | 20 | | 20 | | | | 60 | | | |
| 21,22 | | | | 10 | 10 | | 10 | | 10 | | | | 60 | | | |
| 23,24 | | | | 10 | 10 | | 20 | | | | | | 60 | | | |
| 25 | | | | | | | | | 40 | | | | 60 | | | |
| 26 | | | | | | | 20 | | 20 | | | | 60 | | | |
| 27 | | 20 | | | | | 10 | | 10 | | | | 60 | | | |
| 28 | | 10 | | | | | 15 | | 15 | | | | 60 | | | |
| 29 | 15 | | | | | | 10 | | 15 | | | | 60 | | | |
| 30 | 20 | | | | | | | 10 | 10 | | | | 60 | | | |

**Table A5.** Cross-walking table for ESA-CCI LC class 40 - Mosaic natural vegetation (tree, shrub, herbaceous cover)(>50%) / cropland(<50%)

| | 1 | 2 | 3 | 4 | 5 | 6 | 7 | 8 | 9 | 10 | 11 | 12 | 13 | 14 | 15 | 16 |
|---|---|---|---|---|---|---|---|---|---|---|---|---|---|---|---|---|
| | Tree | | | | | | Shrub | | Grass | | Special vegetation | | Crops | | Non-vegetated | |
| Holdridge Life Zone | tropical broadleaf evergreen | tropical broadleaf deciduous | temperate broadleaf evergreen | temperate broadleaf deciduous | evergreen coniferous | deciduous coniferous | evergreen | deciduous | C3 | C4 | Tundra | Swamps | crops | | urban | bare ground |
| 1,2 | | | | | | | | | | | 35 | 30 | 35 | | | |
| 3-5 | | | | | | | | | | | 30 | 35 | 35 | | | |
| 6 | | | | | | | | | | | 25 | 40 | 35 | | | |
| 7 | | | | | | | | | 60 | | | | 40 | | | |
| 8 | | | | | 10 | | | | 50 | | | | 40 | | | |
| 9,10 | | | | | 15 | | | | 45 | | | | 40 | | | |
| 11 | | | | | 20 | | | | 40 | | | | 40 | | | |
| 12 | | | | | 30 | | | 20 | 10 | | | | 40 | | | |
| 13 | | | | 10 | 10 | | | 10 | 30 | | | | 40 | | | |
| 14,15 | | | | 20 | 20 | | | 10 | 10 | | | | 40 | | | |
| 16 | | | | 25 | 20 | | | | 15 | | | | 40 | | | |
| 17 | | | | 25 | 25 | | | | 10 | | | | 40 | | | |
| 18 | | | | 30 | 30 | | | | | | | | 40 | | | |
| 19 | | | | | | | | | 60 | | | | 40 | | | |
| 20 | | | | | | | 35 | | 25 | | | | 40 | | | |
| 21 | | | 20 | | 15 | | 15 | | 10 | | | | 40 | | | |
| 22 | | | 25 | | 10 | | 15 | | 10 | | | | 40 | | | |
| 23,24 | | | 20 | | 20 | | 20 | | | | | | 40 | | | |
| 25 | | | | | | | | | 60 | | | | 40 | | | |
| 26 | | | | | | | 30 | | 30 | | | | 40 | | | |
| 27 | | 10 | | | | | 50 | | | | | | 40 | | | |
| 28 | | 40 | | | | | 20 | | | | | | 40 | | | |
| 29 | 40 | | | | | | 20 | | | | | | 40 | | | |
| 30 | 50 | | | | | | 10 | | | | | | 40 | | | |





**Table A6.** Cross-walking table for ESA-CCI LC class 50 - Tree cover, broadleaved, evergreen, closed to open (>15%)

| | 1 | 2 | 3 | 4 | 5 | 6 | 7 | 8 | 9 | 10 | 11 | 12 | 13 | 14 | 15 | 16 |
|---|---|---|---|---|---|---|---|---|---|---|---|---|---|---|---|---|
| | Tree | | | | | | Shrub | | Grass | | Special vegetation | | Crops | | Non-vegetated | |
| Holdridge Life Zone | tropical broadleaf evergreen | tropical broadleaf deciduous | temperate broadleaf evergreen | temperate broadleaf deciduous | evergreen coniferous | deciduous coniferous | evergreen | deciduous | C3 | C4 | Tundra | Swamps | crops | | urban | bare ground |
| 1-6 | | | 12.5 | | | | 12.5 | | | | 75 | | | | | |
| 7-18 | | | 90 | 10 | | | | | | | | | | | | |
| 19-24 | | | 100 | | | | | | | | | | | | | |
| 25-30 | 100 | | | | | | | | | | | | | | | |

**Table A7.** Cross-walking table for ESA-CCI LC class 60 - Tree cover, broadleaved, deciduous, closed to open (>15%)

| | 1 | 2 | 3 | 4 | 5 | 6 | 7 | 8 | 9 | 10 | 11 | 12 | 13 | 14 | 15 | 16 |
|---|---|---|---|---|---|---|---|---|---|---|---|---|---|---|---|---|
| | Tree | | | | | | Shrub | | Grass | | Special vegetation | | Crops | | Non-vegetated | |
| Holdridge Life Zone | tropical broadleaf evergreen | tropical broadleaf deciduous | temperate broadleaf evergreen | temperate broadleaf deciduous | evergreen coniferous | deciduous coniferous | evergreen | deciduous | C3 | C4 | Tundra | Swamps | crops | | urban | bare ground |
| 1-6 | | | | | | | | 100 | | | | | | | | |
| 7-24 | | | | 70 | | | | 15 | 15 | | | | | | | |
| 25-30 | | 70 | | | | | | 15 | 15 | | | | | | | |

**Table A8.** Cross-walking table for ESA-CCI LC class 61 - Tree cover, broadleaved, deciduous, closed (>40%)

| | 1 | 2 | 3 | 4 | 5 | 6 | 7 | 8 | 9 | 10 | 11 | 12 | 13 | 14 | 15 | 16 |
|---|---|---|---|---|---|---|---|---|---|---|---|---|---|---|---|---|
| | Tree | | | | | | Shrub | | Grass | | Special vegetation | | Crops | | Non-vegetated | |
| Holdridge Life Zone | tropical broadleaf evergreen | tropical broadleaf deciduous | temperate broadleaf evergreen | temperate broadleaf deciduous | evergreen coniferous | deciduous coniferous | evergreen | deciduous | C3 | C4 | Tundra | Swamps | crops | | urban | bare ground |
| 1-6 | | | | | | | | 85 | 15 | | | | | | | |
| 7-24 | | | | 70 | | | | 15 | 15 | | | | | | | |
| 25-30 | | 70 | | | | | | 15 | 15 | | | | | | | |

**Table A9.** Cross-walking table for ESA-CCI LC class 62 - Tree cover, broadleaved, deciduous, open (15-40%)

| | 1 | 2 | 3 | 4 | 5 | 6 | 7 | 8 | 9 | 10 | 11 | 12 | 13 | 14 | 15 | 16 |
|---|---|---|---|---|---|---|---|---|---|---|---|---|---|---|---|---|
| | Tree | | | | | | Shrub | | Grass | | Special vegetation | | Crops | | Non-vegetated | |
| Holdridge Life Zone | tropical broadleaf evergreen | tropical broadleaf deciduous | temperate broadleaf evergreen | temperate broadleaf deciduous | evergreen coniferous | deciduous coniferous | evergreen | deciduous | C3 | C4 | Tundra | Swamps | crops | | urban | bare ground |
| 1-6 | | | | | | | | 65 | 35 | | | | | | | |
| 7-24 | | | | 30 | | | | 25 | 45 | | | | | | | |
| 25-30 | | 30 | | | | | | 25 | 45 | | | | | | | |



**Table A10.** Cross-walking table for ESA-CCI LC class 70 - Tree cover, needleleaved, evergreen, closed to open (>15%) and LC class 71 - Tree cover, needleleaved, evergreen, closed (>40%)

| | 1 | 2 | 3 | 4 | 5 | 6 | 7 | 8 | 9 | 10 | 11 | 12 | 13 | 14 | 15 | 16 |
|---|---|---|---|---|---|---|---|---|---|---|---|---|---|---|---|---|
| | Tree | | | | | | Shrub | | Grass | | Special vegetation | | Crops | | Non-vegetated | |
| **Holdridge Life Zone** | tropical broadleaf evergreen | tropical broadleaf deciduous | temperate broadleaf evergreen | temperate broadleaf deciduous | evergreen coniferous | deciduous coniferous | evergreen | deciduous | C3 | C4 | Tundra | Swamps | crops | | urban | bare ground |
| 1-6 | | | | | 35 | 35 | 15 | | 15 | | | | | | | |
| 7-18 | | | | | 70 | | 10 | 5 | 15 | | | | | | | |
| 19-24 | | | 35 | | 35 | | 10 | 5 | 15 | | | | | | | |
| 25-30 | | | | | 70 | | 10 | 5 | 15 | | | | | | | |

**Table A11.** Cross-walking table for ESA-CCI LC class 72 - Open (15-40%) needleleaved deciduous or evergreen forest (>5m)

| | 1 | 2 | 3 | 4 | 5 | 6 | 7 | 8 | 9 | 10 | 11 | 12 | 13 | 14 | 15 | 16 |
|---|---|---|---|---|---|---|---|---|---|---|---|---|---|---|---|---|
| | Tree | | | | | | Shrub | | Grass | | Special vegetation | | Crops | | Non-vegetated | |
| **Holdridge Life Zone** | tropical broadleaf evergreen | tropical broadleaf deciduous | temperate broadleaf evergreen | temperate broadleaf deciduous | evergreen coniferous | deciduous coniferous | evergreen | deciduous | C3 | C4 | Tundra | Swamps | crops | | urban | bare ground |
| 1-6 | | | | | 15 | 15 | 25 | | 45 | | | | | | | |
| 7-18 | | | | | 30 | | 20 | 5 | 45 | | | | | | | |
| 19-24 | | | 15 | | 15 | | 20 | 5 | 45 | | | | | | | |
| 25-30 | | | | | 30 | | 20 | 5 | 45 | | | | | | | |

**Table A12.** Cross-walking table for ESA-CCI LC class 80 - Tree cover, needleleaved, deciduous, closed to open (>15%)

| | 1 | 2 | 3 | 4 | 5 | 6 | 7 | 8 | 9 | 10 | 11 | 12 | 13 | 14 | 15 | 16 |
|---|---|---|---|---|---|---|---|---|---|---|---|---|---|---|---|---|
| | Tree | | | | | | Shrub | | Grass | | Special vegetation | | Crops | | Non-vegetated | |
| **Holdridge Life Zone** | tropical broadleaf evergreen | tropical broadleaf deciduous | temperate broadleaf evergreen | temperate broadleaf deciduous | evergreen coniferous | deciduous coniferous | evergreen | deciduous | C3 | C4 | Tundra | Swamps | crops | | urban | bare ground |
| 1-30 | | | | | | 50 | 5 | 15 | 30 | | | | | | | |

**Table A13.** Cross-walking table for ESA-CCI LC class 81 - Treecover, needleleaved, deciduous, closed (>40%)

| | 1 | 2 | 3 | 4 | 5 | 6 | 7 | 8 | 9 | 10 | 11 | 12 | 13 | 14 | 15 | 16 |
|---|---|---|---|---|---|---|---|---|---|---|---|---|---|---|---|---|
| | Tree | | | | | | Shrub | | Grass | | Special vegetation | | Crops | | Non-vegetated | |
| **Holdridge Life Zone** | tropical broadleaf evergreen | tropical broadleaf deciduous | temperate broadleaf evergreen | temperate broadleaf deciduous | evergreen coniferous | deciduous coniferous | evergreen | deciduous | C3 | C4 | Tundra | Swamps | crops | | urban | bare ground |
| 1-30 | | | | | | 70 | | 15 | 15 | | | | | | | |

**Table A14.** Cross-walking table for ESA-CCI LC class 82 - Tree cover, needleleaved, deciduous, open (15-40%)

| | 1 | 2 | 3 | 4 | 5 | 6 | 7 | 8 | 9 | 10 | 11 | 12 | 13 | 14 | 15 | 16 |
|---|---|---|---|---|---|---|---|---|---|---|---|---|---|---|---|---|
| | Tree | | | | | | Shrub | | Grass | | Special vegetation | | Crops | | Non-vegetated | |
| **Holdridge Life Zone** | tropical broadleaf evergreen | tropical broadleaf deciduous | temperate broadleaf evergreen | temperate broadleaf deciduous | evergreen coniferous | deciduous coniferous | evergreen | deciduous | C3 | C4 | Tundra | Swamps | crops | | urban | bare ground |
| 1-30 | | | | | | 30 | 5 | 20 | 45 | | | | | | | |





**Table A15.** Cross-walking table for ESA-CCI LC class 90 - Tree cover, mixed leaf type (broadleaved and needleleaved)

| | 1 | 2 | 3 | 4 | 5 | 6 | 7 | 8 | 9 | 10 | 11 | 12 | 13 | 14 | 15 | 16 |
|---|---|---|---|---|---|---|---|---|---|---|---|---|---|---|---|---|
| | Tree | | | | | | Shrub | | Grass | | Special vegetation | | Crops | | Non-vegetated | |
| Holdridge Life Zone | tropical broadleaf evergreen | tropical broadleaf deciduous | temperate broadleaf evergreen | temperate broadleaf deciduous | evergreen coniferous | deciduous coniferous | evergreen | deciduous | C3 | C4 | Tundra | Swamps | crops | | urban | bare ground |
| 1-12 | | | | 20 | 70 | | | | 10 | | | | | | | |
| 13-24 | | | | 70 | 20 | | | | 10 | | | | | | | |
| 25-30 | 45 | 45 | | | | | | | 10 | | | | | | | |

**Table A16.** Cross-walking table for ESA-CCI LC class 100 - Mosaic tree and shrub (>50%) / herbaceous cover(<50%)

| | 1 | 2 | 3 | 4 | 5 | 6 | 7 | 8 | 9 | 10 | 11 | 12 | 13 | 14 | 15 | 16 |
|---|---|---|---|---|---|---|---|---|---|---|---|---|---|---|---|---|
| | Tree | | | | | | Shrub | | Grass | | Special vegetation | | Crops | | Non-vegetated | |
| Holdridge Life Zone | tropical broadleaf evergreen | tropical broadleaf deciduous | temperate broadleaf evergreen | temperate broadleaf deciduous | evergreen coniferous | deciduous coniferous | evergreen | deciduous | C3 | C4 | Tundra | Swamps | crops | | urban | bare ground |
| 1 | | | | | 30 | | 30 | | | | 30 | 10 | | | | |
| 2,3 | | | | | 30 | | 25 | | | | 25 | 20 | | | | |
| 4-6 | | | | | 30 | | 20 | | | | 20 | 30 | | | | |
| 7-9 | | | | 20 | 20 | | 20 | | 40 | | | | | | | |
| 10 | | | | 25 | 25 | | 20 | | 30 | | | | | | | |
| 11 | | | | 30 | 30 | | 20 | | 20 | | | | | | | |
| 12 | | | | 30 | 30 | | 25 | | 15 | | | | | | | |
| 13 | | | | 15 | 15 | | | 35 | 35 | | | | | | | |
| 14 | | | | 20 | 20 | | | 30 | 30 | | | | | | | |
| 15 | | | | 25 | 25 | | | 25 | 25 | | | | | | | |
| 16-18 | | | | 25 | 25 | | | 30 | 20 | | | | | | | |
| 19,20 | | | 30 | | | | 30 | | 40 | | | | | | | |
| 21,22 | | | 35 | | | | 35 | | 30 | | | | | | | |
| 23,24 | | | 40 | | | | 30 | | 30 | | | | | | | |
| 25 | | 20 | | | | | 50 | | 30 | | | | | | | |
| 26 | | 25 | | | | | 50 | | 25 | | | | | | | |
| 27 | | 30 | | | | | 45 | | 25 | | | | | | | |
| 28 | | 40 | | | | | 35 | | 25 | | | | | | | |
| 29 | | 60 | | | | | 20 | | 20 | | | | | | | |
| 30 | | 70 | | | | | 15 | | 15 | | | | | | | |

**Table A17.** Cross-walking table for ESA-CCI LC class 110 - Mosaic herbaceous cover (>50%) / tree and shrub (<50%)

| | 1 | 2 | 3 | 4 | 5 | 6 | 7 | 8 | 9 | 10 | 11 | 12 | 13 | 14 | 15 | 16 |
|---|---|---|---|---|---|---|---|---|---|---|---|---|---|---|---|---|
| | Tree | | | | | | Shrub | | Grass | | Special vegetation | | Crops | | Non-vegetated | |
| Holdridge Life Zone | tropical broadleaf evergreen | tropical broadleaf deciduous | temperate broadleaf evergreen | temperate broadleaf deciduous | evergreen coniferous | deciduous coniferous | evergreen | deciduous | C3 | C4 | Tundra | Swamps | crops | | urban | bare ground |
| 1-6 | | | | | | | 50 | | | | 45 | 5 | | | | |
| 7 | | | | 10 | 10 | | 20 | | 60 | | | | | | | |
| 8 | | | | 10 | 20 | | 10 | | 60 | | | | | | | |
| 9 | | | | 25 | 25 | | | | 50 | | | | | | | |
| 10 | | | | 30 | 30 | | | | 40 | | | | | | | |
| 11,12 | | | | 35 | 35 | | | | 30 | | | | | | | |
| 13 | | | | 15 | 15 | | | | 70 | | | | | | | |
| 14,15 | | | | 20 | 10 | | | | 70 | | | | | | | |
| 16 | | | | 30 | | | | 10 | 60 | | | | | | | |
| 17,18 | | | | 35 | | | | 15 | 50 | | | | | | | |
| 19 | | | 15 | | | | 15 | | 70 | | | | | | | |
| 20 | | | 10 | | | | 20 | | 70 | | | | | | | |
| 21 | | | 20 | | | | 10 | | 70 | | | | | | | |
| 22 | | | 30 | | | | 10 | | 60 | | | | | | | |
| 23,24 | | | 35 | | | | 15 | | 50 | | | | | | | |
| 25 | | 15 | | | | | 15 | | 70 | | | | | | | |
| 26 | | 20 | | | | | 10 | | 70 | | | | | | | |
| 27 | | 25 | | | | | 15 | | 60 | | | | | | | |
| 28 | | 30 | | | | | | 10 | 60 | | | | | | | |
| 29 | | 40 | | | | | | 10 | 50 | | | | | | | |
| 30 | | 50 | | | | | | 10 | 40 | | | | | | | |



**Table A18.** Cross-walking table for ESA-CCI LC class 120 - Shrubland

| | 1 | 2 | 3 | 4 | 5 | 6 | 7 | 8 | 9 | 10 | 11 | 12 | 13 | 14 | 15 | 16 |
|---|---|---|---|---|---|---|---|---|---|---|---|---|---|---|---|---|
| | Tree | | | | | | Shrub | | Grass | | Special vegetation | | Crops | | Non-vegetated | |
| Holdridge Life Zone | tropical broadleaf evergreen | tropical broadleaf deciduous | temperate broadleaf evergreen | temperate broadleaf deciduous | evergreen coniferous | deciduous coniferous | evergreen | deciduous | C3 | C4 | Tundra | Swamps | crops | | urban | bare ground |
| 1-6 | | | | | | | 40 | | | | 55 | 5 | | | | |
| 7-12 | | | | | | | 10 | 50 | 40 | | | | | | | |
| 13 | | | | | | | 70 | | 30 | | | | | | | |
| 14 | | | | | | | 40 | 30 | 30 | | | | | | | |
| 15 | | | | | | | 20 | 60 | 20 | | | | | | | |
| 16 | | | | | | | 20 | 70 | 10 | | | | | | | |
| 17,18 | | | | | | | 10 | 80 | 10 | | | | | | | |
| 19 | | | | | | | 10 | | 90 | | | | | | | |
| 20 | | | | | | | 50 | | 50 | | | | | | | |
| 21 | | | | | | | 90 | | 10 | | | | | | | |
| 22 | | | | | | | 80 | 10 | 10 | | | | | | | |
| 23,24 | | | | | | | 100 | | | | | | | | | |
| 25 | | | | | | | 10 | 10 | 80 | | | | | | | |
| 26,27 | | | | | | | 20 | 60 | 20 | | | | | | | |
| 28 | | | | | | | 10 | 70 | 20 | | | | | | | |
| 29,30 | | | | | | | 10 | 80 | 10 | | | | | | | |

**Table A19.** Cross-walking table for ESA-CCI LC class 121 - Evergreen shrubland and LC class 122 - Deciduous Shrubland

| | 1 | 2 | 3 | 4 | 5 | 6 | 7 | 8 | 9 | 10 | 11 | 12 | 13 | 14 | 15 | 16 |
|---|---|---|---|---|---|---|---|---|---|---|---|---|---|---|---|---|
| | Tree | | | | | | Shrub | | Grass | | Special vegetation | | Crops | | Non-vegetated | |
| Holdridge Life Zone | tropical broadleaf evergreen | tropical broadleaf deciduous | temperate broadleaf evergreen | temperate broadleaf deciduous | evergreen coniferous | deciduous coniferous | evergreen | deciduous | C3 | C4 | Tundra | Swamps | crops | | urban | bare ground |
| 1-6 | | | | | | | 40 | | | | 55 | 5 | | | | |
| 7-12 | | | | | | | 60 | | 40 | | | | | | | |
| 13,14 | | | | | | | 70 | | 30 | | | | | | | |
| 15 | | | | | | | 80 | | 20 | | | | | | | |
| 16-18 | | | | | | | 90 | | 10 | | | | | | | |
| 19 | | | | | | | 10 | | 90 | | | | | | | |
| 20 | | | | | | | 50 | | 50 | | | | | | | |
| 21,22 | | | | | | | 90 | | 10 | | | | | | | |
| 23,24 | | | | | | | 100 | | | | | | | | | |
| 25 | | | | | | | 20 | | 80 | | | | | | | |
| 26-28 | | | | | | | 80 | | 20 | | | | | | | |
| 29,30 | | | | | | | 90 | | 10 | | | | | | | |

**Table A20.** Cross-walking table for ESA-CCI LC class 122 - Evergreen shrubland and LC class 122 - Deciduous Shrubland

| | 1 | 2 | 3 | 4 | 5 | 6 | 7 | 8 | 9 | 10 | 11 | 12 | 13 | 14 | 15 | 16 |
|---|---|---|---|---|---|---|---|---|---|---|---|---|---|---|---|---|
| | Tree | | | | | | Shrub | | Grass | | Special vegetation | | Crops | | Non-vegetated | |
| Holdridge Life Zone | tropical broadleaf evergreen | tropical broadleaf deciduous | temperate broadleaf evergreen | temperate broadleaf deciduous | evergreen coniferous | deciduous coniferous | evergreen | deciduous | C3 | C4 | Tundra | Swamps | crops | | urban | bare ground |
| 1-6 | | | | | | | 40 | | | | 55 | 5 | | | | |
| 7-12 | | | | | | | | 60 | 40 | | | | | | | |
| 13,14 | | | | | | | | 70 | 30 | | | | | | | |
| 15 | | | | | | | | 80 | 20 | | | | | | | |
| 16-18 | | | | | | | | 90 | 10 | | | | | | | |
| 19 | | | | | | | | 10 | 90 | | | | | | | |
| 20 | | | | | | | | 50 | 50 | | | | | | | |
| 21,22 | | | | | | | | 90 | 10 | | | | | | | |
| 23,24 | | | | | | | | 100 | | | | | | | | |
| 25 | | | | | | | | 80 | 20 | | | | | | | |
| 26-28 | | | | | | | | 80 | 20 | | | | | | | |
| 29,30 | | | | | | | | 90 | 10 | | | | | | | |





**Table A21.** Cross-walking table for ESA-CCI LC class 130 - Grassland

| | 1 | 2 | 3 | 4 | 5 | 6 | 7 | 8 | 9 | 10 | 11 | 12 | 13 | 14 | 15 | 16 |
|---|---|---|---|---|---|---|---|---|---|---|---|---|---|---|---|---|
| | Tree | | | | | | Shrub | | Grass | | Special vegetation | | Crops | | Non-vegetated | |
| Holdridge Life Zone | tropical broadleaf evergreen | tropical broadleaf deciduous | temperate broadleaf evergreen | temperate broadleaf deciduous | evergreen coniferous | deciduous coniferous | evergreen | deciduous | C3 | C4 | Tundra | Swamps | crops | | urban | bare ground |
| 1-6 | | | | | | | | | | | 90 | 10 | | | | |
| 7-13 | | | | | | | | | 100 | | | | | | | |
| 14 | | | | | | | 5 | | 95 | | | | | | | |
| 15 | | | | | | | | 7.5 | 92.5 | | | | | | | |
| 16 | | | | | | | | 10 | 90 | | | | | | | |
| 17 | | | | | | | | 12.5 | 87.5 | | | | | | | |
| 18 | | | | | | | | 15 | 85 | | | | | | | |
| 19 | | | | | | | | | 100 | | | | | | | |
| 20,21 | | | | | | | 5 | | 95 | | | | | | | |
| 22 | | | | | | | 7.5 | | 92.5 | | | | | | | |
| 23,24 | | | | | | | 10 | | 90 | | | | | | | |
| 25 | | | | | | | | | 100 | | | | | | | |
| 26 | | | | | | | 5 | | 95 | | | | | | | |
| 27 | | | | | | | 5 | 5 | 90 | | | | | | | |
| 28 | | | | | | | | 10 | 90 | | | | | | | |
| 29 | | | | | | | | 12.5 | 87.5 | | | | | | | |
| 30 | | | | | | | | 15 | 85 | | | | | | | |

**Table A22.** Cross-walking table for ESA-CCI LC class 140 - Lichens and mosses

| | 1 | 2 | 3 | 4 | 5 | 6 | 7 | 8 | 9 | 10 | 11 | 12 | 13 | 14 | 15 | 16 |
|---|---|---|---|---|---|---|---|---|---|---|---|---|---|---|---|---|
| | Tree | | | | | | Shrub | | Grass | | Special vegetation | | Crops | | Non-vegetated | |
| Holdridge Life Zone | tropical broadleaf evergreen | tropical broadleaf deciduous | temperate broadleaf evergreen | temperate broadleaf deciduous | evergreen coniferous | deciduous coniferous | evergreen | deciduous | C3 | C4 | Tundra | Swamps | crops | | urban | bare ground |
| 1-6 | | | | | | | | | | | 90 | 10 | | | | |
| 7-30 | | | | | | | | | 100 | | | | | | | |





**Table A23.** Cross-walking table for ESA-CCI LC class 150 - Sparse vegetation (tree, shrub, herbaceouscover)(<15%)

| | 1 | 2 | 3 | 4 | 5 | 6 | 7 | 8 | 9 | 10 | 11 | 12 | 13 | 14 | 15 | 16 |
|---|---|---|---|---|---|---|---|---|---|---|---|---|---|---|---|---|
| | Tree | | | | | | Shrub | | Grass | | Special vegetation | | Crops | | Non-vegetated | |
| Holdridge Life Zone | tropical broadleaf evergreen | tropical broadleaf deciduous | temperate broadleaf evergreen | temperate broadleaf deciduous | evergreen coniferous | deciduous coniferous | evergreen | deciduous | C3 | C4 | Tundra | Swamps | crops | | urban | bare ground |
| 1-6 | | | | | | | | | | | 50 | 10 | | | | 40 |
| 7-12 | | | | | | | 10 | | 40 | | | | | | | 50 |
| 13 | | | | 5 | 5 | | 5 | | 35 | | | | | | | 50 |
| 14 | | | | 5 | 5 | | 10 | | 30 | | | | | | | 50 |
| 15 | | | | 5 | 5 | | | 10 | 30 | | | | | | | 50 |
| 16 | | | | 5 | 5 | | | 20 | 20 | | | | | | | 50 |
| 17,18 | | | | 10 | 10 | | | 20 | 10 | | | | | | | 50 |
| 19 | | | | | | | 5 | | 45 | | | | | | | 50 |
| 20,21 | | | 5 | | | | 5 | | 40 | | | | | | | 50 |
| 22 | | 5 | | | | | 10 | | 35 | | | | | | | 50 |
| 23 | | | 10 | | | | 10 | | 30 | | | | | | | 50 |
| 24 | | | 15 | | | | 15 | | 20 | | | | | | | 50 |
| 25 | | | | | | | 5 | 5 | 40 | | | | | | | 50 |
| 26,27 | | 10 | | | | | 5 | 5 | 30 | | | | | | | 50 |
| 28,29 | | 10 | | | | | | 20 | 20 | | | | | | | 50 |
| 30 | 10 | | | | | | | 20 | 20 | | | | | | | 50 |

**Table A24.** Cross-walking table for ESA-CCI LC class 151 - Sparse tree (<15%)

| | 1 | 2 | 3 | 4 | 5 | 6 | 7 | 8 | 9 | 10 | 11 | 12 | 13 | 14 | 15 | 16 |
|---|---|---|---|---|---|---|---|---|---|---|---|---|---|---|---|---|
| | Tree | | | | | | Shrub | | Grass | | Special vegetation | | Crops | | Non-vegetated | |
| Holdridge Life Zone | tropical broadleaf evergreen | tropical broadleaf deciduous | temperate broadleaf evergreen | temperate broadleaf deciduous | evergreen coniferous | deciduous coniferous | evergreen | deciduous | C3 | C4 | Tundra | Swamps | crops | | urban | bare ground |
| 1-6 | | | | | | | | | | | 50 | 10 | | | | 40 |
| 7-12 | | | | | | | 10 | | 40 | | | | | | | 50 |
| 13 | | | | 5 | 5 | | | | 40 | | | | | | | 50 |
| 14,15 | | | | 5 | 10 | | | | 35 | | | | | | | 50 |
| 16 | | | | 10 | 5 | | | | 35 | | | | | | | 50 |
| 17,18 | | | | 10 | 10 | | | | 30 | | | | | | | 50 |
| 19-21 | | | 5 | | | | | | 45 | | | | | | | 50 |
| 22 | | | 10 | | | | | | 40 | | | | | | | 50 |
| 23 | | | 15 | | | | | | 35 | | | | | | | 50 |
| 24 | | | 20 | | | | | | 30 | | | | | | | 50 |
| 25 | | 10 | | | | | | | 40 | | | | | | | 50 |
| 26-29 | | 15 | | | | | | | 35 | | | | | | | 50 |
| 30 | 15 | | | | | | | | 35 | | | | | | | 50 |





**Table A25.** Cross-walking table for ESA-CCI LC class 152 - Sparse shrub (<15%)

| | 1 | 2 | 3 | 4 | 5 | 6 | 7 | 8 | 9 | 10 | 11 | 12 | 13 | 14 | 15 | 16 |
|---|---|---|---|---|---|---|---|---|---|---|---|---|---|---|---|---|
| | Tree | | | | | | Shrub | | Grass | | Special vegetation | | Crops | | Non-vegetated | |
| Holdridge Life Zone | tropical broadleaf evergreen | tropical broadleaf deciduous | temperate broadleaf evergreen | temperate broadleaf deciduous | evergreen coniferous | deciduous coniferous | evergreen | deciduous | C3 | C4 | Tundra | Swamps | crops | | urban | bare ground |
| 1 | | | | | | | 5 | | | | 45 | 10 | | | | 40 |
| 2-6 | | | | | | | 10 | | | | 40 | 10 | | | | 40 |
| 7-10 | | | | | | | 10 | | 40 | | | | | | | 50 |
| 11,12 | | | | | | | 20 | | 30 | | | | | | | 50 |
| 13,14 | | | | | | | 10 | | 40 | | | | | | | 50 |
| 15,16 | | | | | | | | 15 | 35 | | | | | | | 50 |
| 17,18 | | | | | | | | 20 | 30 | | | | | | | 50 |
| 19 | | | | | | | 5 | | 45 | | | | | | | 50 |
| 20,21 | | | | | | | 10 | | 40 | | | | | | | 50 |
| 22,23 | | | | | | | 15 | | 35 | | | | | | | 50 |
| 24 | | | | | | | 20 | | 30 | | | | | | | 50 |
| 25 | | | | | | | 5 | | 45 | | | | | | | 50 |
| 26 | | | | | | | 10 | | 40 | | | | | | | 50 |
| 27 | | | | | | | 7.5 | 7.5 | 35 | | | | | | | 50 |
| 28,29 | | | | | | | | 15 | 35 | | | | | | | 50 |
| 30 | | | | | | | | 20 | 30 | | | | | | | 50 |

**Table A26.** Cross-walking table for ESA-CCI LC class 153 - Sparse herbaceous cover (<15%)

| | 1 | 2 | 3 | 4 | 5 | 6 | 7 | 8 | 9 | 10 | 11 | 12 | 13 | 14 | 15 | 16 |
|---|---|---|---|---|---|---|---|---|---|---|---|---|---|---|---|---|
| | Tree | | | | | | Shrub | | Grass | | Special vegetation | | Crops | | Non-vegetated | |
| Holdridge Life Zone | tropical broadleaf evergreen | tropical broadleaf deciduous | temperate broadleaf evergreen | temperate broadleaf deciduous | evergreen coniferous | deciduous coniferous | evergreen | deciduous | C3 | C4 | Tundra | Swamps | crops | | urban | bare ground |
| 1-6 | | | | | | | | | | | 40 | 10 | | | | 50 |
| 7-30 | | | | | | | | | 50 | | | | | | | 50 |



**Table A27.** Cross-walking table for ESA-CCI LC class 160 - Tree cover, flooded, fresh or brakish water

| | 1 | 2 | 3 | 4 | 5 | 6 | 7 | 8 | 9 | 10 | 11 | 12 | 13 | 14 | 15 | 16 |
|---|---|---|---|---|---|---|---|---|---|---|---|---|---|---|---|---|
| | Tree | | | | | | Shrub | | Grass | | Special vegetation | | Crops | | Non-vegetated | |
| Holdridge Life Zone | tropical broadleaf evergreen | tropical broadleaf deciduous | temperate broadleaf evergreen | temperate broadleaf deciduous | evergreen coniferous | deciduous coniferous | evergreen | deciduous | C3 | C4 | Tundra | Swamps | crops | | urban | bare ground |
| 1-6 | | | | 10 | | | | | | | 45 | 45 | | | | |
| 7-18 | | | | 70 | | | | | | | | 30 | | | | |
| 19-24 | | | 70 | | | | | | | | | 30 | | | | |
| 25-30 | 35 | 35 | | | | | | | | | | 30 | | | | |

**Table A28.** Cross-walking table for ESA-CCI LC class 170 - Tree cover, flooded, saline water

| | 1 | 2 | 3 | 4 | 5 | 6 | 7 | 8 | 9 | 10 | 11 | 12 | 13 | 14 | 15 | 16 |
|---|---|---|---|---|---|---|---|---|---|---|---|---|---|---|---|---|
| | Tree | | | | | | Shrub | | Grass | | Special vegetation | | Crops | | Non-vegetated | |
| Holdridge Life Zone | tropical broadleaf evergreen | tropical broadleaf deciduous | temperate broadleaf evergreen | temperate broadleaf deciduous | evergreen coniferous | deciduous coniferous | evergreen | deciduous | C3 | C4 | Tundra | Swamps | crops | | urban | bare ground |
| 1-6 | | | | | 40 | | 30 | | | | 10 | 20 | | | | |
| 7-12 | | | | 20 | | | | 60 | | | | 20 | | | | |
| 13-18 | | | | 30 | | | | 50 | | | | 20 | | | | |
| 19-24 | | | 60 | | | | 10 | 10 | | | | 20 | | | | |
| 25-30 | 80 | | | | | | | | | | | 20 | | | | |





**Table A29.** Cross-walking table for ESA-CCI LC class 180 - Shrub or herbaceous cover, flooded, fresh / saline / brakish water

| Holdridge Life Zone | 1 Tree tropical broadleaf evergreen | 2 tropical broadleaf deciduous | 3 temperate broadleaf evergreen | 4 temperate broadleaf deciduous | 5 evergreen coniferous | 6 deciduous coniferous | 7 Shrub evergreen | 8 deciduous | 9 Grass C3 | 10 C4 | 11 Special vegetation Tundra | 12 Swamps | 13 Crops crops | 14 | 15 Non-vegetated urban | 16 bare ground |
|---|---|---|---|---|---|---|---|---|---|---|---|---|---|---|---|---|
| 1-6 | | | | | | | | | | | 95 | 5 | | | | |
| 7 | | | | | | | | 10 | | | | 90 | | | | |
| 8 | | | | | | | 15 | 15 | 20 | | | 50 | | | | |
| 9 | | | | | | | 20 | 20 | 20 | | | 40 | | | | |
| 10-12 | | | | | | | 20 | 20 | 20 | | | 40 | | | | |
| 13 | | | | | | | | 20 | 20 | | | 60 | | | | |
| 14 | | | | | | | | 25 | 25 | | | 50 | | | | |
| 15 | | | | | | | | 30 | 30 | | | 40 | | | | |
| 16 | | | | | | | | 35 | 35 | | | 30 | | | | |
| 17,18 | | | | | | | | 45 | 15 | | | 40 | | | | |
| 19,20 | | | | | | | 30 | 40 | 30 | | | | | | | |
| 21,22 | | | | | | | 40 | 40 | 20 | | | | | | | |
| 23 | | | | | | | 40 | 50 | 10 | | | | | | | |
| 24 | | | | | | | 30 | 60 | 10 | | | | | | | |
| 25 | | | | | | | 30 | 30 | 40 | | | | | | | |
| 26 | | | | | | | 30 | 40 | 30 | | | | | | | |
| 27 | | | | | | | 40 | 40 | 20 | | | | | | | |
| 28 | | | | | | | 40 | 50 | 10 | | | | | | | |
| 29 | | | | | | | 70 | 30 | | | | | | | | |
| 30 | | | | | | | 90 | 10 | | | | | | | | |

**Table A30.** Cross-walking table for ESA-CCI LC class 190 - Urban

| Holdridge Life Zone | 1 Tree tropical broadleaf evergreen | 2 tropical broadleaf deciduous | 3 temperate broadleaf evergreen | 4 temperate broadleaf deciduous | 5 evergreen coniferous | 6 deciduous coniferous | 7 Shrub evergreen | 8 deciduous | 9 Grass C3 | 10 C4 | 11 Special vegetation Tundra | 12 Swamps | 13 Crops crops | 14 | 15 Non-vegetated urban | 16 bare ground |
|---|---|---|---|---|---|---|---|---|---|---|---|---|---|---|---|---|
| 1-30 | | | | | | | | | | | | | | | 100 | |

**Table A31.** Cross-walking table for ESA-CCI LC class 200 - Bare areas, LC class 201 - Consolidated bare areas and LC class 202 - Unconsolidated bare areas.

| Holdridge Life Zone | 1 Tree tropical broadleaf evergreen | 2 tropical broadleaf deciduous | 3 temperate broadleaf evergreen | 4 temperate broadleaf deciduous | 5 evergreen coniferous | 6 deciduous coniferous | 7 Shrub evergreen | 8 deciduous | 9 Grass C3 | 10 C4 | 11 Special vegetation Tundra | 12 Swamps | 13 Crops crops | 14 | 15 Non-vegetated urban | 16 bare ground |
|---|---|---|---|---|---|---|---|---|---|---|---|---|---|---|---|---|
| 1-30 | | | | | | | | | | | | | | | | 100 |





## Appendix B

**Figure B1.** Total count of GT-SUR points per 2.5° grid cell (a-c; g-i) and producer's accuracy for the individual LULC groups (d-f;j-l) for filter set 5 (dominant LULC group occupies > 50% per LANDMATE PFT grid cell)

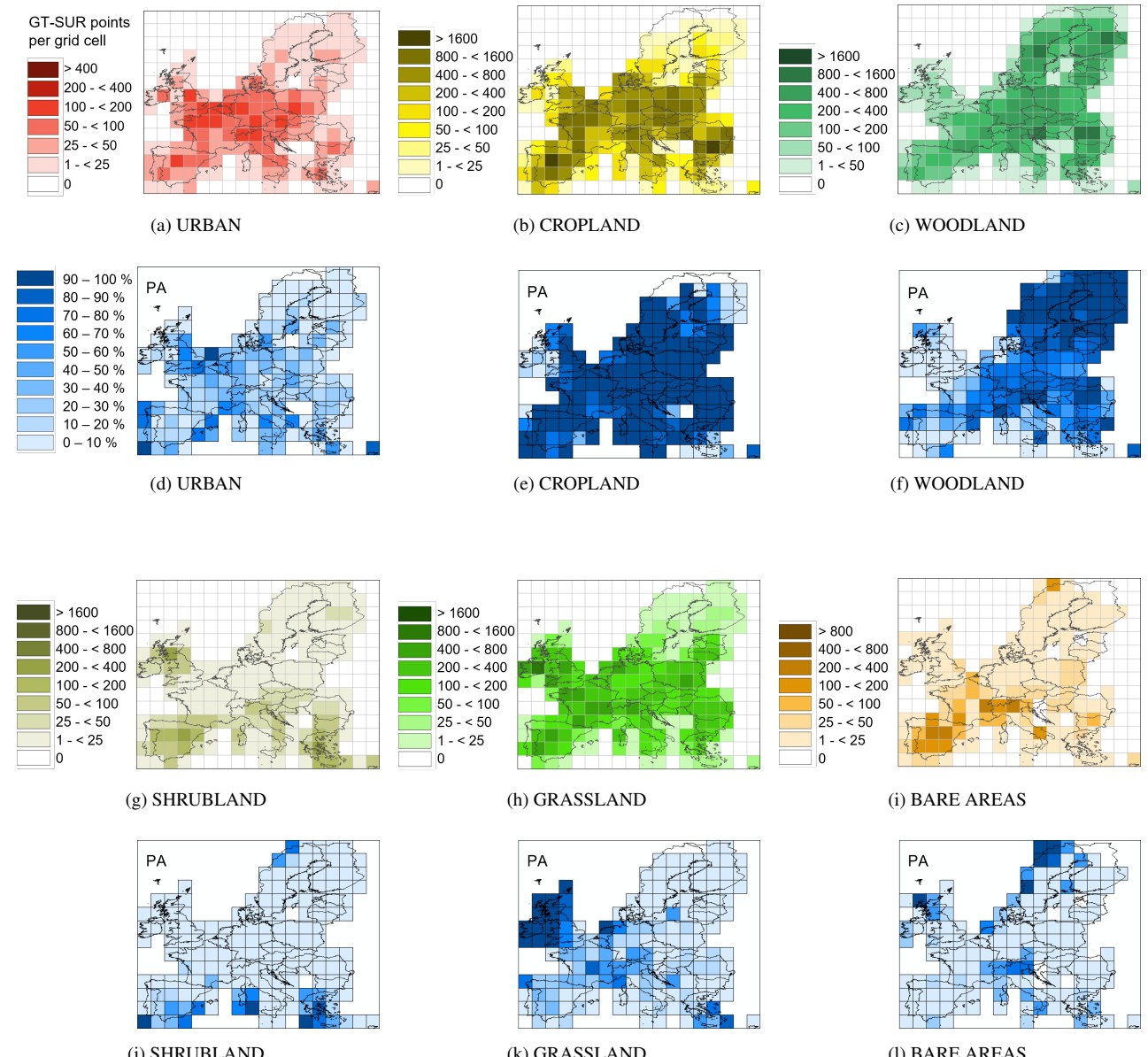

**Figure B2.** Total count of GT-SUR points per 2.5° grid cell (a-c; g-i) and producer's accuracy for the individual LULC groups (d-f;j-l) for filter set 7 (dominant LULC group occupies > 70% per LANDMATE PFT grid cell)

|   | 1 | 2 | 3 | 4 | 5 | 6 | 7 | SUM | UA |
|---|---|---|---|---|---|---|---|---|---|
| 1 | 3234 | 806 | 1063 | 178 | 1769 | 120 | 407 | 7577 | 42.68 |
| 2 | 6625 | 67374 | 22298 | 5444 | 28559 | 4185 | 2485 | 136970 | 49.19 |
| 3 | 2414 | 5081 | 88064 | 8989 | 12818 | 1527 | 5544 | 124437 | 70.77 |
| 4 | 624 | 5316 | 4637 | 5498 | 1789 | 439 | 1487 | 19790 | 27.78 |
| 5 | 1411 | 4515 | 8063 | 6082 | 20763 | 1767 | 1643 | 44244 | 46.93 |
| 6 | 82 | 199 | 200 | 830 | 567 | 1810 | 460 | 4148 | 43.64 |
| 7 | 3 | 4 | 49 | 277 | 276 | 530 | 314 | 1453 | 21.61 |
| SUM | 14393 | 83295 | 124374 | 27298 | 66541 | 10378 | 12340 | | |
| PA | 22.47 | 80.887 | 70.81 | 20.14 | 31.20 | 17.44 | 2.54 | OA: | 55.24 |

**Table B1.** Confusion matrix for LANDMATE PFT filter set 1 - Dominant LULC group occupies a minimum of 10 % of a LANDMATE PFT grid cell

|   | 1 | 2 | 3 | 4 | 5 | 6 | 7 | SUM | UA |
|---|---|---|---|---|---|---|---|---|---|
| 1 | 3234 | 806 | 1063 | 178 | 1769 | 120 | 407 | 7577 | 42.68 |
| 2 | 6625 | 67374 | 22298 | 5444 | 28559 | 4185 | 2485 | 136970 | 49.19 |
| 3 | 2414 | 5081 | 88064 | 8989 | 12818 | 1527 | 5544 | 124437 | 70.77 |
| 4 | 624 | 5316 | 4637 | 5498 | 1789 | 439 | 1487 | 19790 | 27.78 |
| 5 | 1411 | 4515 | 8063 | 6082 | 20763 | 1767 | 1643 | 44244 | 46.93 |
| 6 | 82 | 199 | 200 | 830 | 567 | 1810 | 460 | 4148 | 43.64 |
| 7 | 3 | 4 | 49 | 277 | 276 | 530 | 314 | 1453 | 21.61 |
| SUM | 14393 | 83295 | 124374 | 27298 | 66541 | 10378 | 12340 | | |
| PA | 22.47 | 80.887 | 70.81 | 20.14 | 31.20 | 17.44 | 2.54 | OA: | 55.24 |

**Table B2.** Confusion matrix for LANDMATE PFT filter set 2 - Dominant LULC group occupies a minimum of 20 % of a LANDMATE PFT grid cell





|  | 1 | 2 | 3 | 4 | 5 | 6 | 7 | SUM | UA |
|---|---|---|---|---|---|---|---|---|---|
| 1 | 3221 | 793 | 1041 | 174 | 1748 | 117 | 404 | 7498 | 42.96 |
| 2 | 6596 | 67323 | 22210 | 5395 | 28488 | 4168 | 2457 | 136637 | 49.27 |
| 3 | 2377 | 5034 | 87838 | 8903 | 12750 | 1511 | 5483 | 123896 | 70.90 |
| 4 | 615 | 5280 | 4484 | 5363 | 1748 | 425 | 1401 | 19316 | 27.76 |
| 5 | 1401 | 4485 | 7961 | 5983 | 20716 | 1754 | 1559 | 43859 | 47.23 |
| 6 | 78 | 187 | 186 | 798 | 552 | 1799 | 452 | 4052 | 44.40 |
| 7 | 3 | 4 | 47 | 276 | 275 | 530 | 310 | 1445 | 21.45 |
| SUM | 14291 | 83106 | 123767 | 26892 | 66277 | 10304 | 12066 |  |  |
| PA | 22.54 | 81.01 | 70.97 | 19.94 | 31.26 | 17.46 | 2.57 | OA: | 55.41 |

**Table B3.** Confusion matrix for LANDMATE PFT filter set 3 - Dominant LULC group occupies a minimum of 30 % of a LANDMATE PFT grid cell

|  | 1 | 2 | 3 | 4 | 5 | 6 | 7 | SUM | UA |
|---|---|---|---|---|---|---|---|---|---|
| 1 | 3079 | 715 | 904 | 152 | 1597 | 109 | 364 | 6920 | 44.49 |
| 2 | 6263 | 66184 | 20069 | 4795 | 27209 | 4034 | 2304 | 130858 | 50.58 |
| 3 | 2061 | 4045 | 83073 | 7509 | 11168 | 1274 | 5030 | 114160 | 72.77 |
| 4 | 501 | 4813 | 3013 | 4235 | 1392 | 329 | 742 | 15025 | 28.19 |
| 5 | 1238 | 4031 | 6748 | 5091 | 19572 | 1571 | 1219 | 39470 | 49.59 |
| 6 | 54 | 123 | 122 | 606 | 469 | 1681 | 425 | 3480 | 48.30 |
| 7 | 2 | 2 | 40 | 254 | 258 | 517 | 252 | 1325 | 19.02 |
| SUM | 13198 | 79913 | 113969 | 22642 | 61665 | 9515 | 10336 |  |  |
| PA | 23.33 | 82.82 | 72.89 | 18.70 | 31.74 | 17.67 | 2.44 | OA: | 57.22 |

**Table B4.** Confusion matrix for LANDMATE PFT filter set 4 - Dominant LULC group occupies a minimum of 40 % of a LANDMATE PFT grid cell



|  | 1 | 2 | 3 | 4 | 5 | 6 | 7 | SUM | UA |
|---|---|---|---|---|---|---|---|---|---|
| 1 | 2632 | 499 | 676 | 117 | 1218 | 84 | 292 | 5518 | 47.70 |
| 2 | 5482 | 62499 | 15269 | 3772 | 23519 | 3737 | 1913 | 116191 | 53.79 |
| 3 | 1510 | 2215 | 71799 | 5277 | 7767 | 853 | 4284 | 93705 | 76.62 |
| 4 | 362 | 3865 | 1752 | 2689 | 915 | 206 | 350 | 10139 | 26.52 |
| 5 | 933 | 2992 | 4373 | 3605 | 16306 | 1227 | 893 | 30329 | 53.76 |
| 6 | 31 | 61 | 62 | 292 | 321 | 1375 | 392 | 2534 | 54.26 |
| 7 | 1 | 0 | 29 | 110 | 214 | 233 | 70 | 657 | 10.65 |
| SUM | 10951 | 72131 | 93960 | 15862 | 50260 | 7715 | 8194 |  |  |
| PA | 24.03 | 86.65 | 76.41 | 16.95 | 32.44 | 17.82 | 0.85 | OA: | 60.74 |

**Table B5.** Confusion matrix for LANDMATE PFT filter set 5 - Dominant LULC group occupies a minimum of 50 % of a LANDMATE PFT grid cell

|  | 1 | 2 | 3 | 4 | 5 | 6 | 7 | SUM | UA |
|---|---|---|---|---|---|---|---|---|---|
| 1 | 2123 | 284 | 464 | 85 | 844 | 67 | 231 | 4098 | 51.81 |
| 2 | 4436 | 56963 | 10802 | 2887 | 19016 | 3314 | 1556 | 98974 | 57.55 |
| 3 | 1025 | 978 | 57212 | 2949 | 4699 | 488 | 3345 | 70696 | 80.93 |
| 4 | 194 | 2459 | 967 | 1713 | 518 | 122 | 240 | 6213 | 27.57 |
| 5 | 628 | 1847 | 2584 | 2333 | 12497 | 798 | 630 | 21317 | 58.62 |
| 6 | 14 | 27 | 34 | 104 | 181 | 1022 | 339 | 1721 | 59.38 |
| 7 | 1 | 0 | 18 | 40 | 153 | 87 | 25 | 324 | 7.72 |
| SUM | 8421 | 62558 | 72081 | 10111 | 37908 | 5898 | 6366 |  |  |
| PA | 25.21 | 91.06 | 79.37 | 16.94 | 32.97 | 17.33 | 0.39 | OA: | 64.70 |

**Table B6.** Confusion matrix for LANDMATE PFT filter set 6 - Dominant LULC group occupies a minimum of 60 % of a LANDMATE PFT grid cell



|     | 1    | 2     | 3     | 4    | 5     | 6    | 7    | SUM   | UA    |
|-----|------|-------|-------|------|-------|------|------|-------|-------|
| 1   | 1684 | 167   | 311   | 53   | 568   | 44   | 185  | 3012  | 55.91 |
| 2   | 3288 | 49624 | 7217  | 2088 | 14351 | 2840 | 1145 | 80553 | 61.60 |
| 3   | 414  | 255   | 30158 | 806  | 1745  | 177  | 1910 | 35465 | 85.04 |
| 4   | 40   | 793   | 458   | 988  | 191   | 42   | 160  | 2672  | 36.98 |
| 5   | 410  | 1053  | 1363  | 1415 | 9113  | 478  | 425  | 14257 | 63.92 |
| 6   | 5    | 11    | 15    | 61   | 104   | 768  | 302  | 1266  | 60.66 |
| 7   | 1    | 0     | 9     | 19   | 99    | 50   | 9    | 187   | 4.81  |
| SUM | 5842 | 51903 | 39531 | 5430 | 26171 | 4399 | 4136 |       |       |
| PA  | 28.83| 95.61 | 76.29 | 18.20| 34.82 | 17.46| 0.22 | OA:   | 67.20 |

**Table B7.** Confusion matrix for LANDMATE PFT filter set 7 - Dominant LULC group occupies a minimum of 70 % of a LANDMATE PFT grid cell

|     | 1    | 2     | 3    | 4    | 5     | 6    | 7    | SUM   | UA    |
|-----|------|-------|------|------|-------|------|------|-------|-------|
| 1   | 1261 | 83    | 208  | 29   | 369   | 32   | 138  | 2120  | 59.48 |
| 2   | 2009 | 38997 | 4002 | 1296 | 9321  | 2239 | 745  | 58609 | 66.54 |
| 3   | 32   | 21    | 3201 | 54   | 195   | 8    | 108  | 3619  | 88.45 |
| 4   | 10   | 74    | 198  | 442  | 51    | 9    | 106  | 890   | 49.66 |
| 5   | 241  | 518   | 640  | 691  | 5957  | 240  | 229  | 8516  | 69.95 |
| 6   | 3    | 5     | 10   | 39   | 62    | 533  | 268  | 920   | 57.93 |
| 7   | 1    | 0     | 6    | 8    | 53    | 17   | 6    | 91    | 6.59  |
| SUM | 3557 | 39698 | 8265 | 2559 | 16008 | 3078 | 1600 |       |       |
| PA  | 35.45| 98.23 | 38.73| 17.27| 37.21 | 17.32| 0.38 | OA:   | 67.41 |

**Table B8.** Confusion matrix for LANDMATE PFT filter set 8 - Dominant LULC group occupies a minimum of 80 % of a LANDMATE PFT grid cell



| | 1 | 2 | 3 | 4 | 5 | 6 | 7 | SUM | UA |
|---|---|---|---|---|---|---|---|---|---|
| 1 | 808 | 44 | 111 | 14 | 207 | 16 | 89 | 1289 | 62.68 |
| 2 | 592 | 17167 | 877 | 414 | 2601 | 1043 | 269 | 22963 | 74.76 |
| 3 | 1 | 1 | 47 | 1 | 1 | 0 | 14 | 65 | 72.31 |
| 4 | 2 | 7 | 28 | 74 | 11 | 1 | 10 | 133 | 55.64 |
| 5 | 40 | 81 | 108 | 181 | 1358 | 83 | 58 | 1909 | 71.14 |
| 6 | 3 | 1 | 7 | 20 | 28 | 338 | 230 | 627 | 53.91 |
| 7 | 0 | 0 | 1 | 2 | 2 | 1 | 1 | 7 | 14.29 |
| SUM | 1446 | 17301 | 1179 | 706 | 4208 | 1482 | 671 | | |
| PA | 55.88 | 99.23 | 3.99 | 10.48 | 32.27 | 22.81 | 0.15 | OA: | 73.33 |

**Table B9.** Confusion matrix for LANDMATE PFT filter set 9 - Dominant LULC group occupies a minimum of 90 % of a LANDMATE PFT grid cell

| | 1 | 2 | 3 | 4 | 5 | 6 | 7 | SUM | UA |
|---|---|---|---|---|---|---|---|---|---|
| 1 | 252 | 10 | 28 | 0 | 40 | 8 | 51 | 389 | 64.78 |
| 2 | 22 | 565 | 16 | 7 | 52 | 14 | 20 | 696 | 81.18 |
| 3 | 0 | 0 | 0 | 0 | 0 | 0 | 0 | 0 | / |
| 4 | 0 | 0 | 0 | 0 | 0 | 0 | 0 | 0 | / |
| 5 | 0 | 1 | 4 | 14 | 48 | 6 | 1 | 74 | 64.86 |
| 6 | 2 | 0 | 4 | 7 | 9 | 112 | 156 | 290 | 38.62 |
| 7 | 0 | 0 | 0 | 0 | 0 | 0 | 0 | 0 | / |
| SUM | 276 | 576 | 52 | 28 | 149 | 140 | 228 | | |
| PA | 91.30 | 98.09 | 0.00 | 0.00 | 32.21 | 80.00 | 0.00 | OA: | 67.43 |

**Table B10.** Confusion matrix for LANDMATE PFT filter set 10 - Dominant LULC group occupies 100 % of a LANDMATE PFT grid cell



*Author contributions.* VR conceptualized the paper outline and objective with the support of DR, PH and JB. VR and PH developed the
540 cross-walking procedure and the corresponding cross-walking tables. PH developed the required translation software for the cross-walking
procedure. VR developed the accuracy assessment design for the LANDMATE PFT map supported by BB. VR conducted the accuracy
assessment and the visualization of results. VR wrote the original draft of the paper, VR, PH, DR, JB, and BB reviewed and edited the draft.
VR wrote the final paper

*Competing interests.* The authors declare that they have no conflict of interest.

*Acknowledgements.* This work was financed within the framework of the Helmholtz Institute for Climate Service Science (HICSS), a co-
operation between Climate Service Center Germany (GERICS) and Universität Hamburg, Germany and conducted as part of the project
LANDMATE (Modelling human LAND surface modifications and its feedbacks on local and regional cliMATE). We acknowledge the sup-
port of LUCAS by WCRP-CORDEX as a Flagship Pilot Study. We acknowledge the E-OBS dataset from the EU-FP6 project UERRA
(http://www.uerra.eu) and the Copernicus Climate Change Service, and the data providers in the ECA&D project (https://www.ecad.eu).
We thank the European Space Agency (ESA) for making the Land cover products publicly available. Special thanks go to the FPS LUCAS
partners for providing useful comments in order to improve the dataset.





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
