# Peer review of "High-resolution land-use land-cover change data for regional climate modelling applications over Europe - Part 1: The plant functional type basemap for 2015"

_Earth System Science Data, 2021_

## Author Comment (AC1)

**Author Response to the Reviewer Comments to the manuscript "High-resolution land-use land-cover change data for regional climate modelling applications over Europe – Part 1: The plant functional type basemap for 2015" [essd-2021-251] submitted to Earth System and Science Data.**

We would kindly thank the Editor David Carlson for coordinating the Review Process as well as Arthur Endsley and one anonymous referee for their reviews. These have been very useful for improving the dataset as well as the manuscript. The individual comments are listed below (shown in blue) including our responses (shown in black). The changes discussed in this reply will be included in the revised manuscript and dataset and will thus become visible after re-submission.

**Response to Reviewer 1**

**General Comments**

One first issue is based on the style. The document does not read super well, partly because it is being very descriptive on a case by case basis with respect to the classes. The text is too long in this respect and many of the details are not so relevant. The results were each class in analyzed separately with respect to filtering (e.g. figures 6, 8, 10-14 and related text) are very tedious. I suggest this should be condensed in some form or another, if not removed and relegated to annexes. There are other issues too that make the reading difficult (not in terms of language, but style), some of which I detail in the remarks further below.

Thank you for this general suggestion to prepare the results more concise. We revised the section and merged the individual graphs in a way that provides a quick overview for the reader in the beginning of the section. Pleas see an example below.

[Figure]

Another major issue I have is that the validation based on GT-SUR is done only on the LANDMATE PFT map, and not on the baseline ESA CCI LC map, nor on the ESA CCI LC map transformant to PFT with the standard CWT. This makes it impossible to judge on the improvement in quality provided by the present study. It may very well be that all the accuracy with respect to GT-SUR is from the correct classification of the LC in the first place, and not at all by the added value provided by the study. I suppose this is not the case, but it is not proven by the study. In fact, readers might think the authors are misleading the audience by claiming accuracy when none is due. To defend themselves

from this, I strongly suggest that the authors engage in making the same comparison with GT-SUR with some baseline reference before they apply their contribution (and I would suggest to do it both for the LC, and for the PFTs with the default CWT). In that way, they will be able to isolate and quantify the added-value of their contribution

Thank you for pointing this out. It is in fact extremely important that the validation is also done to the ESA CCI PFT to show the improvement achieved by the CWT modification and the additional data used within the LANDMATE PFT workflow. We applied the validation worflow to ESA CCI PFT (prepared with the standard CWT) and added a section within the results, where the agreement with GT-SUR of both PFT datasets is compared.

Another point that I think needs improvement is the whole part on the 'filtering'. It is not fully clear to me what is being done or why. It needs to be explained better and probably simplified. Also, all this part (along with the results) occupy a very substantial part of the manuscript and it is not fully clear why. Also, why are things aggregated at 2.5 dd instead of done at the 0.1dd? I fully understand that there are mismatches between the samples at point level and grid level, but I am not convinced by what is done here. I probably misunderstood something, but again, it should be crystal clear why things are done

The filtering is done in order to investigate the dependence of the accuracy metrics to the grid cell heterogeneity. With the filtering we create groups of a minimum coverage of the dominant LULC type and get a range of accuracy measures for these groups, which is more reasonable than calculating the metrics only for the dominant LULC type, regardless their actual coverage.

For visualization of the spatial results we tried different ways. Due to the high point count within the research area, some kind of aggregation needed to be done to be able to interpret the results. We tested different aggregation resolutions for the auxiliary grid to achieve such an aggregation and we found that the 2.5° grid is most suitable. Nevertheless, the accuracy metrics are calculated based on the individual LANDMATE PFT grid cells.

Last major point is that I do not see why the dataset is done only for 2015. Especially since the ESA CCI LC maps are done for 1992-2020 now, and that they had been designed to be consistent in time. I think most users would expect you to exploit this info and provide not just a single date basemap but also a set of consistent maps for the observation period.

Thank you for pointing this out. The production, validation and publication of a LANDMATE PFT time series is planned as future work in the LANDMATE project. However, the aim of the LANDMATE PFT map is to provide a basemap for a long-term dataset, which provides both historical as well as projected LULCC (i.e., the LUCAS LUC dataset, which is described in the companion paper).

**Specific Comments**

(L7) "to achieve the *best* possible representation"? Arguing it is the best is perhaps exaggerating. Unless you prove that cannot be anything better.

Thank you for this comment. The wording should indeed be changed. We eliminated the word "best" out of the sentence to not mislead the reader.

(L31) I think it may be misleading that the concept of PFTs in increasingly used. It has been used for a long while now, and it is unclear if there is a trend in using it more. Some models are perhaps drifting away from it, or at least evolving in a bit of a different way… see: https://doi.org/10.1111/gcb.13910 and https://doi.org/10.1029/2018MS001453

Thank you for this comment. It is true that the PFT concept is only one of the trends in the climate modelling community. Yet, for regional climate modelling, the PFT concept is used in many model families, such as the RCMs coupled to CLM (e.g. COSMO-CLM, RegCM-CLM, WRF-CLM) and REMO-iMOVE (see for instance Davin et al. 2020). We adjusted the sentence and added relevant references to emphasize the meaning.

(L38) as it seems clear the CWPs will be a key concept in the rest of the paper, it would be good to describe here a bit more what it is. The more general reader would not necessarily understand why a specific "procedure" is needed for "translating" LULC to PFT. I would further recommend to spend more time explaining the fundamental difference between LULC and PFTs. Thirdly, I would go forward and cite studies that have actually employed a CWP in practice (https://doi.org/10.1038/s41467-021-24551-5 and https://doi.org/10.1038/s41467-017-02810-8 come to mind, there must be others). A separate dedicated paragraph in the intro on all of this would be warranted.

Thank you for this comment. Indeed, it is very important to understand why a CWP is needed and how a CWP works. Some of the comments from reviewer 2 are also related to this issue. Hence, we added a paragraph that explains our CWP using an example class. In addition, we extented the discription of the PFT concept in the introduction, explained the differences to the remotely sensed LULC classes from ESA CCI LC and cited the suggested literature.

(L48) is it specific for the RCM community? Could it not be applied more generally to the ESM (Earth System Modelling) community, irrespective of whether it is region or global?

The transfer of the LANDMATE PFT map and the corresponding procedure to the LULC input for Earth System Models would require the consideration of more factors in the climate system, such as the carbon cycle that is currently not accounted for in RCMs. Therefore, LANDMATE PFT or the corresponding time series LUCAS LUC (companion paper) is tailored for the use in RCMs. However, it could be potentially modified to work in ESMs, which is out of the scope of the present work.

(L75) While I understand the logic of having a general workflow sub-section first to present an overview, the way it is it actually complicates the understanding. For instance, you talk about the HLZ map (L84) without saying before what it is (or rather describing what it is). It feels that maybe this section is not so useful as it is… if you want to keep it, it should be made much more generic so that it does not depend on what you will explain after.

Thank you for pointing out that the use of the dataset names might be confusing to the reader. However, we see it beneficial for the understanding of the specific features of LANDMATE PFT to put the general workflow in the beginning. We modified the text in a way that gives the reader an overview without knowing the acronyms of the datasets used. The acronyms are added to the respective cross-reference (in brackets) that leads to the section describing the respective dataset.

Further, we added a sentence to the involvement of the potential C3/C4 grass map within the workflow within the section 2.1 "General workflow".

(L87) Not clear here (even if it may become clearer after) why an individual CWT is needed for each LULC class. The idea of a CWT would (a priori) be that you have one table for all.

Thank you for pointing out the confusing wording here. The CWT used here is one large table for the whole dataset but through the involvment of the HLZs, the table/matrix has three dimensions (LULC,HLZ,PFT). For reasons of transparency and for easier visualization for the reader the table was

split up for each LC class respectively. The text should make this clear now. In addition, we added one CWT from the appendix into the section for the reader to get a quick look at the resulting CWTs.

(L117) This simplified downscaling by bilinear interpolation will probably lead to problems in mountain areas. Any way to mitigate this? Could you comment on the possible consequences?

Thank you for this question. The EOBS dataset has a 0.1° resolution. Hence, the issue of interpolation of the atmospheric data is rather small for the generation of the LANDMATE PFT 0.1° map. For the LANDMATE PFT 0.018° map the bilinear interpolation cannot account for the small-scale variation of temperature and rainfall. In order to resolve this issue higher resolution climate data or more sophisticated downscaling, especially of rainfall, would be needed. Downscaling of EOBS data would also introduce uncertainties and needs to be checked very carefully and we wanted to use the same dataset for the both LANDMATE PFT resolutions for consistency reasons.

In addition, when we were looking into this issue we found an interpolation issue that caused in fact problems near coastlines in Ireland as well as on the western coastline of Norway. The method of Hoffmann et al. (2016), which takes care of the extrapolation of values at coastlines, was not correctly implemented leaving coastal grid points with incorrect values. However, the issue only occurs for the LANDMATE PFT map in 0.018° horizontal resolution while the 0.1° map is only hardly affected. The validation procedure and the resulting accuracy figures are also only slightly affected because there are only a few ground truth observations near the coast.

While LANDMATE PFT version 1.0 is already published, the issue will be addressed in the following version 1.1 that is already being prepared. For this manuscript, the issue and the affected regions are addressed in the discussion section with reference to the preparation of LANDMATE PFT Version 1.1.

Typo was corrected

(Figure 2) There a logical inconsistency in the colour gradients you use. To be consistent, please always keep "deserts" as either the lighter or darker colour in each scale

Thank you for pointing out the incorrect visualization in the Holdridge Life Zone Map. Within the visualization process, the Holdridge Class numbers were mixed up and this resulted in the wrongly colored figure. We redid the visualization using the correct Holdridge Zone numbering and updated figure 2. In addition, the HLZ numbers from the CWTs in the appendix were added to this figure.

(Table 3) is a bit pointless as a table, as it only has one field (the number is not very meaningful in the context of this paper) and thus is just a list. If you keep this as a table, could you please add more fields with info (e.g. description)

Thank you for pointing this out. The table is indeed a bit pointless. The intention was to introduce the PFT names and corresponding numbers that are later needed to read the CWTs. We resolved this issue as follows. The table was removed. Instead, the PFTs shown in figure 3 get an extended legend where the corresponding numbers are added to the LANDMATE PFT legend. Figure 3 is repositioned in order to have the LANDMATE PFTs shown at an earlier point of the manuscript.

(L149) It is not clear enough where the information to make the C3/C4 distinction comes from. This separation is very important for modelers, yet the CCI LC product (and standard CWT) does not provide any way to do the distinction. This suggests it is difficult to do. Claiming that here you can do it would require more details on how it is done and some validation that is it done correctly. Furthermore,

could you also clarify in what ways this is better than using the Still maps commonly used by modelers (e.g. https://doi.org/10.1029/2001GB001807)

Thank you for pointing this out. Since the ESA CCI LC does not contain C3 or C4 grass land cover classes and since it is not possible to make this distinction using the HLZs, we used the potential C4 grass dataset provided by the North American Carbon Program Multi-scale Synthesis and Terrestrial Model Intercomparison Project. For the creation of this dataset the method of Still et al. (2003) is used but with more recent climate data. We added more information on the creation of the dataset including a citation of the Still et al. (2003) method!

(L174) I don't quite follow. You say six, but the title says 8. Not directly clear why you say "at the expense of two shrub-PFTs". Probably better to rephrase (this and other parts of the paragraph) by being more exhaustive at first (i.e. there are 6 tree classes and 2 shrubs classes. The reason why is … )

The sentence refers to the previous sentence, where the ESA-CCI PFT classification is mentioned. That classification has four shrub-PFTs, a number that was decreased in favor of creating two more tree PFTs. We revised the sentence in order to clarify that.

(L185) typo in class 61

We added the missing letter to the class 61

(L192) Again, the C4/C3 is not clear enough to me, from the text itself. Furthermore, is what Wei et al. did really an accepted way?

As mentioned in the comments related to L149 the method of Still et al. (2003) is used for the creation of the dataset. Hence, we think that this is a credible dataset.

(L200) you have not clearly stated what the actual "translation" is?

Thank you for pointing out the inconsistency in the wording. We replaced the word "translation" with "CWP"

(L202) specify that this is only for the European domain considered here

Thank you for pointing this out. We added the domain of interest to the sentence to make it clear to the reader

(L205) what about C4 crops? According to the authors, the main point of the study is to make a map that reveals traits more than LC, but the C4 vs C3 is perhaps one of the most important ones for vegetation models. So one would expect a proper C4 crop PFT.

The distinction of C3/C4 grass was done on the request of the RCM modelers because of the biogeophysical properties of both grass types. The current RCMs do not consider the carbon- nore the nitrogen cycle, yet. Hence, the biogeochemical differences were seen as less important for the RCM community. While there are of course differences between C3 and C4 crops that are relevant for biogeophysical processes they were, however, regarded as less important than the actual cropland management practices such as irrigation. The latter, has direct and indirect impacts on the surface energy balance and water cycle and is of particular interest of the FPS LUCAS. Hence, it was decided to provide two cropland classes (i.e. irrigated and non-irrigated cropland). In the future, it is possible to add more cropland classes, if they are required and used in regional climate modelling community, by employing additional datasets.

(L210) more info on what the actual translation means would be needed

Thank you again for pointing out the inconsistent wording. We replaced the word "translation" through "CWP"

Thank you for pointing this out. We made it now clear in the text that we found considerable differences between the ESA-CCI LC irrigation class and the FAO irrigation map. That is why we decided not to rely on one of these products for representation of irrigated areas. We did not show this comparison since it was part of the preliminary studies and not of the actual scope of this work

(L223) I have the impression that the same could have been decided for C4 plants. It seems that the level of scrutiny in this irrigated class is not the same as the one applied for C4/C3. In both cases, an external product could be used, but in C4/C3 it was and not for irrigated.

see comments related to L217. In order to achieve consistency for the time series LUCAS LUC (companion paper), it was decided to rely on the irrigated areas given by LUH2. Therefore, the distinction between irrigated and non-irrigated cropland areas is done in the LUCAS LUC workflow, which is presented in the companion paper.

(L229) Are bare classes fully 'non-vegetated', or do they contain residual vegetation? If so, any idea of how much?

Thank you for this question. Within the ESA CCI LC data the bare area classes are non-vegetated. Sparse vegetation patches that have a sufficiently large cover to carry the features of the respective vegetation are represented through the sparse or mosaic vegetation classes. In LANDMATE PFT, patches of vegetation are represented through fractions of the respective vegetation PFT within the same grid cell. The urban type could potentially contain urban green (e.g. street trees, small parks). However, this can be highly variable from city to city and some urban parameterizations are accounting for urban green within an urban class. Hence, the urban grid cells from ESA-CCI LC are directly translated into urban fractions in the LANDMATE PFTs.

(L232-236) I do not see what LANDMATE PFT brings specifically here. The distinction between urban and bare is done by the underlying land cover map, so here the credit goes to the LC cover map and not the present study.

Thank you for this note. The urban land cover proportions from ESA-CCI LC are directly converted to the corresponding LANDMATE PFT "urban". The bare ground fraction originates from bare ground ESA-CCI LC class and from the sparse vegetation classes. Therefore, the overall proportion of the bare ground fraction is determined by the CWP and not just by the underlying ESA-CCI LC map. Further, the distinction between urban and bare areas is important for high-resolution RCM simulations.

(L239) what about inland water, i.e. rivers and small lakes? Are these also set as NA? are the RCM land-sea-mask covering such areas?

Thank you for this question. All types of water, such as inland water and streams are set to no data. We mentioned the water types that are considered as no data in the text to make this clear to the reader.

(L282-285) why bring this issue of cost and size here? Here you have a very large amount of records for free… (ie. You do not need to collect them), so these phrases seem irrelevant.

Thank you for this question. In order to address the whole readership of ESSD, who might not be aware of the components and the challenges associated with an accuracy assessment we mention the specific challenges when it comes to the sampling design and the availability of reference data. We want to emphasize that the case is very rare that one can rely on such a database like the LUCAS ground truth as a reference. We revised the sentences in order to clarify the meaning for the reader

(Table 6) I do not understand what these numbers represent exactly. Samples? Correct samples (i.e. where GT-SUR and LANDMATE match)?

Thank you for pointing this out. The table should give an overview of the proportions of LULC types in GT-SUR and LANDMATE PFT respectively in order to see where there are obvious mismatches regarding the total count. Further, the table should inform the reader about the ratio between total representation and representation with dominant coverage of a LULC type within the LANDMATE PFT cells. We adjusted the table description in the text as well as the describing footnotes to make it more clear to the reader why this table is important.

(Table 7) why not put this as a graph instead of a table? It would be more informative with a single glance

Thank you for this suggestion. In fact, a graph shows the information about the grouping by threshold for minimum coverage in a more simplified way.

(Figure 6) (and others): personally, I find the shadow of the points distractive.

The presentation of results was restructured completely and with that, the shadows in the graphs were removed.

Thank you for pointing out that we used the German spelling here accidentally. We corrected the city name

(General) Format of results is not optimal for understanding. IT should be done in a more condensed way, and not class by class, with a large plot for each. Try to be more synthetic.

Thank you for this comment. We restructured the section completely in order to be more concise with the presentation of results.

(L447) Why in LANDMATE PFT only made available for Europe 2015? The ESA CCI LC is available from 1992 to 2020. The transition from LC to PFT seems relatively straightforward. Why would you not produce the time series? The users would surely expect it.

Please see answer to general comment 4

(L450) this link does not lead anywhere…  at least for me…

Thank you for this hint, the link should possibly lead to the DKRZ download page for our data. We revisited the homepage and updated the link so that now it should work

(L457) "For each ESA-CCI land cover class, an individual CWT is developed…". This does not make sense to me like. The standard way to use the CCI CWT is that it applies weights to each land cover class, and the CWT is the matrix of these weights (rows = LC, cols = PFT). If you apply a CWT to each LC class (to each row), you are just setting the weights for that class.  Now I understand that what you

are actually doing is to set an extra dimension, that of the HLZ. So you need to be more specific, and say that the CWT is decomposed by different HLZ for every LC.

Thank you for addressing this. Please see our answer to your comment (L87) regarding this matter. The wording has been adjusted accordingly

**Response to Reviewer 2**

**General Comments**

Overall, the paper seems to represent an advance in land-cover/ plant functional type (PFT) modeling that informs regional climate models (RCM). I think the paper could be improved by adding more detail about the cross-walk procedure and, in particular, providing a rich example of how the linkage to the Holdridge Life Zones (HLZ) allows for future updates to the land-cover classification in response to environmental change. I gather that, in an RCM, climate changes that push a location out of one HLZ and into another would prompt an update to the corresponding PFT at that location; i.e., the vegetation canopy would be allowed to change and therefore the RCM model state would change in advance of the next time step. This is very interesting and potentially powerful, given that it allows some of the complexity of a dynamic vegetation model to be represented more simply, but it should be made more explicit.

Thank you very much for this positive feedback and the constructive comments. With the additional explanation on the cross-walking procedure we could improve the readability noticeably. The idea that the Holdridge Zones could be computed dynamically is a very good idea for further development and application of this concept. However, the scope of this work was the documentation of the cross-walking procedure and the involvement of additional data as well as the validation of LANDMATE PFT.

There are also a couple of key technical issues with the paper. The crosswalk procedure should be explained in much more detail; the tables in Appendix A are deceptively rich in detail but they really don't make sense without more explication in Section 3. I also think Figure 2 may have the colors for each HLZ mislabeled, as some of the map regions' displayed HLZ classes do not make sense (e.g., desert in Scotland). There are also additional analyses referred to by the authors (ca. Line 487, "The additional comparison with a high-resolution dataset (WSF2015) showed that not only large but also small agglomerations of urban areas are represented well in LANDMATE PFT.") that are completely undescribed in the paper.

The key issues mentioned in this general comment were revised and modified according to the specifications given in the more detailed specific comments below.

**Specific Comments**

(L12-13) The authors write: "A suitable evaluation method has been developed and applied to assess the quality of the new PFT dataset." This is a little vague. Perhaps the authors could be more specific about the type of evaluation method, or remove this sentence from the Abstract?

Thank you very much for this hint. Since the evaluation method is specified in the sentence berfore, we removed the sentence from the abstract.

(L88-95 & APPA) I find the crosswalk tables in Appendix A, and the crosswalk procedure as described, to be confusing. The row sums for every crosswalk table in Appendix A are 100, which seems to imply they are percentages. However, using Table A4 as an example, it's not clear to me

how the ESA-CCI LC class (number 30 in this example) is converted to the LANDMATE PFT based on the HLZ. Typically, a cross-walk table would describe how two uniquely identified, intersecting spatial units are resolved into mutually conflicting classifications. But neither the HLZ nor the ESA-CCI LC class are uniquely identified (i.e., two widely separated locations might have the same HLZ). Is this a probabilistic approach? If the pixel with ESA-CCI LC class 30 is in Holdridge Life Zones 1-6, it has a 20% chance of being classified as Tundra, 20 percent change of being classified as Swamp, and 60 percent chance of being classified as Crops? The authors should be much more explicit about this somewhere in Section 3.

Thank you for this comment. Indeed, it is very important to understand why a CWP is needed and how a CWP works. Some of the comments from reviewer 2 are also related to this issue. Hence, we added a paragraph that explains our CWP using an example class. The CWT used here is one large table for the whole dataset but through the involvement of the HLZs, the table/matrix has three dimensions (LULC,HLZ,PFT). For reasons of transparency and for easier visualization for the reader the table was split up for each LC class respectively. The text should make this clear now. In addition, we added one CWT from the appendix into the section for the reader to get a quick look at the resulting CWTs.

With respect to your example ESA-CCI LC class 30 - Mosaic cropland (>50%) / natural vegetation (tree, shrub, herbaceous cover)(<50%). Let's assume that a given point has the HLZ 5. In this case the fraction of the ESA-CCI LC class 30 is split up into PFTs tundra (20%), swamps (20%), and crops (60%). The percentages are not probabilities but proportions.

(L93-94) The authors write: "This revision of the CWTs is supported by reference data and visual satellite image interpretation." Is this described somewhere? Please add a reference to the appropriate section of the paper (Section 3?).

Thank you for this question. During the modification process of the CWTs the authors looked at the intermediate results and compared the maps to reference data such as CORINE and google earth images, specifically for small regions throughout Europe, where expert knowledge is present. Since there is no quantification for that we decided to eliminate that sentence from the manuscript.

(Figure 2) I think the colors may not be labeled correctly... It seems that much of Italy and Spain are "Rain Forest" and "Wet Forest" variously in the "Warm Temperate" or "Boreal" climate zones while much of Scotland and Ireland appear to be "Desert scrub" or Desert" (Warm temperate climate zone).

Thank you very much for pointing this issue out. It turned out that there was an error in the labeling procedure of the map. Since the issue only occurs within the visualization workflow used to produce the HLZ map, the wrong labeling did not affect the overall classification of LANDMATE PFT 2015. The HLZ map is replaced by a correct version within the overleaf manuscript.

Within this course another confusion could be cleared. The merged Holdridge Zones are in fact the "Warm temperate" and the "Subtropical" Zones and not the "Tropical" and "Subtropical" Zones. The corresponding explanation in the text as well as the legend to figure 2 are corrected.

(Section 3) I think the authors should add a table of their final PFT classification. It is confusing to have Table 3, which is specific to the validation dataset, but not a table of the PFT classification for this new data product. For instance, on Line 204, a reference is made to Table 3, which is the validation dataset's PFT table, during a discussion of the new product's PFT classification.

Thank you for this suggestion. We resolved this issue by eliminating table 3 and moving the figure showing the LANDMATE PFT distribution over Europe to subsection 2.2.5. The figure now includes the numbers and distribution of the LANDMATE PFT.

(L174) Related to the previous comment... The authors write that the target set of PFT classes for trees and shrubs "was done at the expense of two shrub-PFTs." I interpret "at the expense of" as meaning that something was sacrificed, i.e., removed. I look at Figure 3 and see that there are, in fact, two shrub PFTs. Perhaps the authors could revise this sentence?

The sentence refers to the previous sentence, where the ESA-CCI PFT classification is mentioned. That classification has four shrub-PFTs, a number that was decreased in favor of creating two more tree PFTs. We revised the sentence in order to clarify that.

(L315) I think the inline equation is missing an equal sign, i.e., the symbol "$n_{i+}$" should be followed by an equal sign.

Thank you for pointing this out. The equation was modified accordingly.

(L487-488) The authors write: "The additional comparison with a high-resolution dataset (WSF2015) showed that not only large but also small agglomerations of urban areas are represented well in LANDMATE PFT." This is the first time this analysis and this dataset (WSF2015) are mentioned. The authors should describe this analysis as part of the paper (Section 4).

Thank you for pointing this out. The comparison to another dataset (WSF2015) to the LULC type URBAN of LANDMATE PFT is taken out of the analysis since we cannot provide quantification. We emphasized in the text that the fact that we adopted the urban proportions from ESA CCI without modifications and that the structural difference between the reference and the LANDMATE PFT dataset caused the low agreement for the urban representation in the research area.

**All suggested technical corrections were implemented:**

(L20) "was declared an" should be "were declared"

(L30) "as realistic" should be "as realistically"

(L127) "are mereged" should be "are merged"

(L185) "lass 61" should probably be "Class 61"

(L216) A space is missing between the end of one sentence ("2005).") and the start of the next: "Although the ESA-CCI LC..."

(L428) I think Figure 7g needs to be reference along with Figure 7i.

(L489) "and despite of the" should probably be "and despite"

---

## Author Response (AR2)

**Author Response to the Reviewer Comments to the manuscript "High-resolution land-use land-cover change data for regional climate modelling applications over Europe – Part 1: The plant functional type basemap for 2015" [essd-2021-251] submitted to Earth System and Science Data.**

We would kindly thank the Editor David Carlson and the whole editorial team for being highly responsive and helpful throughout the review process. Further, we thank Editor David Carlson and one anonymous referee for their additional reviews. These have been very useful for improving the manuscript, especially the visualization of the results. The individual comments are listed below (shown in blue) including our responses (shown in black). The changes discussed in this reply will be included in the revised manuscript and will thus become visible after re-submission.

**Additional changes**

As decided together with the Editor David Carlson, the companion paper to this publication (essd-2021-252) was rejected. Therefore, we decided to change the title of essd-2021-251 to have it labelled as a standalone publication.

Former title:

**High-resolution land-use land-cover change data for regional climate modelling applications over Europe – Part 1: The plant functional type basemap for 2015**

Updated title:

**High-resolution land-use land-cover dataset for regional climate modelling: A plant functional type map for Europe 2015**

Other changes include the correction of typos or missing punctuation, which are fully documented in the track-changes document submitted together with this supplement.

**Response to Reviewer 1**

The manuscript has been improved. I think there are two outstanding issues: 1) The description of the cross-walking tables (CWTs) and the cross-walking procedure (CWP) still needs improvement (see comment below regarding Line 95 in revised manuscript); 2) The validation approach needs to be more fully described.

Regarding validation and the accuracy of the transformed PFT map... The authors wrote in their response: "During the modification process of the CWTs the authors looked at the intermediate results and compared the maps to reference data such as CORINE and google earth images, specifically for small regions throughout Europe, where expert knowledge is present. Since there is no quantification for that we decided to eliminate that sentence from the manuscript." I don't agree that this information

should be omitted simply because it is a qualitative assessment. The authors should include this in the manuscript as part of their discussion of their validation approach ().

Thank you for this comment. We included a paragraph on the qualitative assessment in the discussion where we clearly state that the efforts done by us were non-quantifiable but that the conduction of a qualitative assessment within the map development process is strongly recommended.

I am satisfied with the validation approach overall. Reviewer 1 is right to point out the problem of proceeding with validation using a grid-based overlay with a grain size 25 times larger (2.5 degrees) than the product under evaluation (0.1 degrees). However, as the authors indicate, "the accuracy metrics are calculated based on the individual LANDMATE PFT grid cells." However, the authors should make this clear in the manuscript (not just the response to reviewers). For example, Line 370 might be updated to distinguish clearly between the 2.5-degree grid used (only) for visualization and the 0.1-degree grid used for accuracy assessment. Furthermore, the various figures showing User's Accuracy and Producer's Accuracy as a function of the minimum dominant PFT fraction should have their captions updated on this point.

Thank you for pointing this out. We added a paragraph with more explanation to section 4.1 "research area" where the 2.5° auxiliary grid is mentioned and shown (Fig.4) for the first time.

One important thing to mention is that for the validation of LANDMATE PFT, the map was produced and used in 0.018° resolution to match the resolution of GT-SUR. To make that clearer to the reader we added this information again in the beginning of the results. We also added a reference to Appendix B where the results (confusion matrices) are available on LANDMATE PFT cell-level.

Table 1 is formatted with very small text. I think this is going to need to be a horizontal/ landscape, full-page table.

Thank you for pointing that out. The table layout is adjusted and in addition, steps were taken to make the table more user friendly (according also to suggestions by other reviewer).

Line 95 in revised manuscript: "For each HLZ in the first column, the LC class 40 is translated into varying fractions of the LANDMATE PFTs." I still do not understand why there are "varying fractions." It seems to me that, for a given pixel, there must be exactly one PFT class. The example CWT and discussion here is a slight improvement over the original manuscript, but the exact use of the CWT is still too vague. I would appreciate it if the authors could extend this example one step further... If I had a GPP model calibrated for LANDMATE PFTs, for this pixel with LC class 40 in HLZ 1 or 2, would I model GPP as the *weighted sum* of GPP in Tundra (35%), Swamps (30%), and Cropland (PFT 13) (35%) canopy? I believe that a concrete example of the *use* of the fractional PFTs, such as this, is essential for conveying how the CWTs are to be used.

Thank you for this comment. We tried to be clearer in the explanation of the CWTs and added a reference to Wilhelm et. al (2014) where the implementation and use of PFTs in the Regional Climate Model REMO is described in detail.

**Response to Reviewer 2**

The manuscript is considerably better than the first draft. There are still several issues that should be considered in my opinion.

paper does not clearly articulate how PFTs are partly seen as outdated for some. Indeed, some models are increasingly turning towards trait-based characterizations of the land instead of using PFTs (see https://doi.org/10.1073/pnas.1304551110 for an example, among others). It is true that current RCMs still use PFTs and thus the present work has value for them, but at the same time I think it is worth mentioning more some of the caveats about PFTs and the future directions that are currently undertaken.

Thank you for emphasizing this. We added a paragraph in the introduction citing relevant sources, where we mention the Plant Functional Traits as parallel concept. However, the Plant Functional Traits are rather to be used in the global climat modelling community. Davin et al. (2020) show clearly that RCM families with a big global user community, such as CLM are employing PFTypes. This citation should help to make it clear to the reader why the LANDMATE PFT map is highly valuable for the RCM community.

Following some of my inquiries about C3/C4 traits the authors state that carbon is not prioritized because RCMs do not model C. However, in Line 35 it is clearly stated that the functional groups are based on both biophysical and biochemical properties. I think this choice of not focusing so much on the carbon should be stated more clearly up-front in the intro/objectives. This should also be reinforced in section 2.2.6. A warning that the C4 trait is inherited from other data, and that it is not evaluated, should be made somewhere in the discussion. Note that it could have been evaluated, as the LUCAS / GT-SUR dataset used does provide crop type, and maize could be used to evaluate if C4 is well caught (as it is the main C4 in Europe).

Thank you for this comment. We clearly state now that the focus of the dataset is to support the RCMs to account for the biophysical processes both in the introduction and in Section 2.2.6. We also state in the discussion that the C3 and C4 grass separation was not evaluated because LUCAS / GT-SUR does not provide this information. Thank you further for pointing put the possibility to evaluate C4 crops using maize as a proxy (at least in Europe). However, in LANDMATE PFT we do not distinguish between different crop types.

L96: missing parenthesis

The parenthesis before the word "Mosaic" was there accidentally and was replaced by a hyphen (according to the class nomenclature by ESA-CCI

Table 1: Make it more user friendly. This could involve (according to me) increasing the fonts, using a colour background for the table cells with numbers, with the intensity of the colour proportional to the magnitude of the number, and perhaps the colour based on the type (Tree, Grass, Crops, etc...). The types of veg could be aligned vertically to allow shrinking the column size. This should then be harmonized with all the tables in the end of the document.

Thank you for pointing this out. The table was modified to be more user friendly. The table was turned sideways, and the font was enlarged as much as possible to still fit on one page. All tables were modified into a common layout using gray row colors every second row starting after the header.

Figure 2: really perturbing choice of colours: Deserts in Cool Temp, have the darkest shade of green. Also, as mentioned in the first review, why are colour intensities in Tropical and Warm going in the opposite of those in cool and boreal? this is not very intuitive... and probably not ideal for colourblind people.

Thank you for this comment. We updated the color scale in order to be more user friendly. The color intensities are going from dry (light) to wet (dark) climates. In addition, the individual color scales for the superior climate zones (e.g. warm temperate, subpolar…) were exchanged to better stand out next to each other.  A colorblind check was done to ensure clear distinction between the HLZs on the map. In this course, some minor mishaps with the numbering of the HLZs on the map was discovered and straightened out.

Figure 3: Grasslands should not be in green to avoid confusion with Trees. The hues in the 6 classes of trees are already too similar and are hard enough to find/identify on the map. The figure caption needs to include an explanation of why the "cropland irrigated" class mentioned as "empty". This is important as it strikes the eye as a major error in the map, as people will expect to see some irrigation dominate areas as in the Po Valley in Italy, or in the Indus and Ganges valley in South Asia.

We updated the color scale to make it easier for the user to identify the individual PFTs. Grassland and trees are kept in green shades but way more distinguished than in the initial figure. A colorblind check was done to ensure clear distinction between the PFTs on the map.

It is correct, that the empty irrigation fractions need to be mentioned within the figure caption. Therefore, a reference to the cropland PFT section 3.4 has been added, where the missing irrigated cropland fractions are addressed in detail.

Section 3.1: this addition of 2 extra tree species boils down to separating "Temperate" from "Tropical" trees. This is thus only added the "tropical" trees beyond the ESA CCI LC, which for Europe is probably VERY marginal (Madeira?). For biophysical effects in Europe (Bright et al) one would expect you make the distinction of Boreal trees. Please comment on this. And perhaps acknowledge the amount to "tropical" trees that are in Europe and that benefit from this increase in PFTs. It is welcome to increase tree PFTs to satisfy the models, but it seems you are not increasing the important type of trees (for Europe that is).

There are tropical trees in the European Domain (which is quite large and also covers part of Africa, Middle east etc.). You are right that a distinction between temperate and boreal tree PFTs could be done as well. We based our CWP on previous work from Wilhelm et al. (2014), who distinguished between the 6 tree PFTs, which are now PFT classes in LANDMATE PFT. If there is a future need for more tree species, it is possible to further refine the CWP distinction could be added in future versions but requires a considerable amount work. We added a paragraph on this issue to the discussion.

L225: missing table ref.

Thank you for pointing this out. The table was excluded from the manuscript within the last review round. The reference points now to the LANDMATE PFT figure where the PFT names and numbers are included.

The justification of not using some irrigation because datasets don't agree fully is a bit weak. This argument could be used also for C4/C3 and for Urban/Bare. At the minimum, this should be re-stated as a caveat in the discussion.

Thank you, you are right. There is additional uncertainty held by the external C4 datasets. However, urban areas are taken from ESA-CCI LC directly and bare ground were also derived from the ESA-CCI LC via the CWP. We included a short paragraph in the discussion on the uncertainties of C4 datasets and that irrigation could be included in future LANDMATE PFT releases.

L262: typo...

The typo was erased.

Figure 5 and 7: remove zeros in y axis by saying counts are in thousands. Also, Fig 5 could be avoided as it can be readily inferred from Fig 7.

See comment below

Figures 8 and 9 should be panels a) and b) of the same figure, one above the other. (Maybe even combind with Fig 7, which would further avoid wasting space with the legend shown 3 times)

Thank you for pointing that out. All changes were done according to reviewers' suggestions. All graphs containing aggregated accuracy metrics for the groups 0.1-1.0 are now panels of fig. 6. Former fig. 5 was eliminated from the manuscript.

Section 5.1: good that this new section was made, as it more clearly provides an idea of where the LANDMATE PFTs might be better than the baseline. It would be good to articulate a bit why the PA for CROPLAND is lower in LANDMATE.

Thank you for pointing this out. Yes, this should indeed be addressed. We added an extra paragraph in section 5.1 where we elaborate on the reason for the worse PA for cropland, which is mainly the translation of cropland types in the Mediterranean region.

L505: "extremely" is probably an exaggeration here, especially since the ESA-CCI provided the CWT approach to deal with this in the first place.

Thank you for this important hint. We modified the sentence and added information in order to acknowledge the default PFT translation given by ESA-CCI.